# Some novel concepts of interval-valued q-rung orthopair fuzzy graphs and computational framework of fuzzy air conditioning system

**Waheed Ahmad Khan**[1*‡], **Sagheer Abbas**[2*‡], **Akhlaq Ahmed**[1‡], **Madhumangal Pal**[3‡], **Muhammad Asif**[4‡], **Muhammad Saeed Khan**[5‡]

**1** Division of Science and Technology, Department of Mathematics, University of Education, Lahore, Punjab, Pakistan, **2** Department of Computer Science, Prince Mohammad Bin Fahd University, Al Khobar, Dhahran, Saudi Arabia, **3** Department of Applied Mathematics with Oceanology and Computer Programming, Vidyasagar University, Midnapore, India, **4** Division of Science and Technology, Department of Computer Science, University of Education, Lahore, Punjab, Pakistan, **5** Department of Electrical, Electronics and Computer Systems, College of Engineering and Technology, University of Sargodha, Sargodha, Pakistan

‡ These authors contributed equally to this work.
* sabbas@pmu.edu.sa (SA); sirwak2003@yahoo.com (WAK)

**Data availability statement:** All relevant data are within the manuscript.

## Abstract

The interval-valued q-rung orthopair fuzzy sets being an extension of interval-valued intuitionistic and interval-valued Pythagorean fuzzy sets is more flexible model to address vague information that has only two attributes yes or no. The combination of the concept of graph structures with interval-valued q-rung orthopair fuzzy sets termed interval-valued q-rung orthopair fuzzy graphs has been introduced in the literature. Due to its nature, interval-valued q-rung orthopair fuzzy graph provides a vast space for membership and non-membership values. In this study, we initiate the notions of covering and matching in the paradigm of interval-valued q-rung orthopair fuzzy graphs (IVq-ROFGs) and provide the analysis of fuzzy air conditioning system (FACS) based on these concepts. Some results and theorems related to these concepts, previously established for different fuzzy graphs, are also extended. In the beginning, we introduce the idea of covering in IVq-ROFGs and investigate several characteristics of some special types of IVq-ROFGs like cyclic IVq-ROFGs, complete IVq-ROFGs and complete bipartite IVq-ROFGs. Afterwards, we introduce the concept of matching in IVq-ROFGs and discuss various types of matchings within this frame. To demonstrate the effectiveness of our study, we apply the concepts of SAs and SIS in IVq-ROFGs supported by algorithm and pseudocode, to establish an appropriate framework for the FACS. This framework assesses the relationship between room temperature and target temperature. Finally, comparative study is conducted to verify that the presented model is the extension of the existing models in the literature.

**Funding:** The author(s) received no specific funding for this work.

**Competing interests:** The authors have declared that no competing interests exist.

# 1 Introduction

In 1965, Zadeh [34] first introduced the term fuzzy sets (FSs). It has become a useful tool to handle the problems with uncertainties. FSs assigns a degree of membership to each element in a given set $S$ from $[0, 1]$. Due to its flexibility, various generalizations of FSs have been introduced. In this context, the concept of interval-valued fuzzy sets (IVFSs) was also explored by Zadeh [35]. The degree of membership in IVFSs was the sub-interval of $[0, 1]$ instead of a single value from the interval $[0, 1]$. IVFSs has a large range for degree of membership and hence more flexible. Recently, numerous concepts such as (m,n)-Fuzzy sets [37], $(a, b)$-Fuzzy soft sets [38], T-spherical FSs [39] in the theory of FSs have been added. The concept of intuitionistic fuzzy sets (IFSs) was presented by Atanassov [2]. In IFSs, degree of membership $(\omega)$ and degree of non-membership $(\nu)$ were allocated to each member under the condition $0 \leq \omega + \nu \leq 1$. However, if we assume the degree of membership is 0.5 and the degree of non-membership is 0.6, then we observe that the condition for IFSs violates. To overcome these difficulties, the notion of PyFSs was introduced by Yager [33]. In PyFSs, the condition for IFSs $0 \leq \omega + \nu \leq 1$ was replaced with $0 \leq (\omega)^2 + (\nu)^2 \leq 1$. Consequently, the above values have been adjusted. The more generalized form of IFSs termed interval-valued intuitionistic fuzzy sets (IVIFSs) was initiated by Atanassov [3]. In IVIFSs, every member has a degree of membership and degree of non-membership in the form of sub-intervals of $[0, 1]$ like $[\omega_L, \omega_U], [\nu_L, \nu_U]$, respectively with the condition $0 \leq \omega_U + \nu_U \leq 1$. For example, the degree of membership and degree of non-membership for IVIFSs are $[0.1, 0.7]$ and $[0.1, 0.2]$, respectively. However, if we consider the degree of non-membership $[0.1, 0.6]$ instead of $[0.1, 0.2]$ in IVIFSs, then again the condition for IVIFSs violates and we are unable to deal with such circumstances through IVIFSs. To handle such situations, the term interval-valued Pythagorean fuzzy sets (IVPyFSs) was proposed by Garg [11]. We observe in the above case of IVIFSs that the values $[0.1, 0.7]$ and $[0.1, 0.6]$ are problematic but have settled in the domain of IVPyFSs which exhibits the condition $0 \leq (0.7)^2 + (0.6)^2 \leq 1$. On the other hand, if we consider the degree of non-membership $[0.3, 0.8]$ instead of $[0.1, 0.6]$ in IVPyFSs, then IVPyFSs fails to handle it. To deal with such obstacles, Peng et al. [20] introduced the concept of q-ROFSs, in which the sum of degree of membership $(\omega)$ and degree of non-membership $(\nu)$ such that $0 \leq (\omega)^q + (\nu)^q \leq 1$ also lies in $[0, 1]$, where $(q \geq 3)$. Consequently, q-ROFSs is comparatively more appropriate and adaptable for the uncertain data. Many researches have been conducted in the realm of q-ROFSs and its numerous applications were explored, such as the distance measure for q-ROFSs and its application in MCDM by Wang et al. [32], a q-ROFSs extension of the DEMATEL and its application in the education sector by Revalde et al. [26], the knowledge measure for the q-ROFSs and its application towards MADM introduced by Khan et al. [17] etc. As an extension of q-ROFSs, the concept of IVq-ROFSs was introduced by Joshi et al. [14]. IVq-ROFSs is most flexible to deal with complex real-world problems as compared to q-ROFSs. Due to its nature, numerous studies have been established on IVq-ROFSs, such as the new possibility degree measure for IVq-ROFSs and its application towards decision-making (DM) by Garg [9], a NGDM method based on CoCoSo and IVq-ROFSs and its application towards MAGDM by Zheng et al. [36], new exponential operation laws and operators for IVq-ROFSs and their application in group DM process by Grag [10]. Moreover, we can appropriately deal with the scenario described in the above example through IVq-ROFSs. By considering the degree of non-membership is $[0.4, 0.8]$, the basic condition of IVq-ROFSs gives us a precise result i.e., $0 \leq (0.7)^q + (0.9)^q \leq 1$, where $(q \geq 3)$. In this way, we can handle many real-world problems by using IVq-ROFSs, more accurately. For more on IVq-ROFSs, one may consult [8].

On the other hand, the idea of fuzzy graphs (FGs) was proposed by Rosenfeld [25]. Similar to the case of FSs, we can easily model any relational phenomenon with uncertain information through FGs. Numerous terms have been described in the paradigm of FGs like fuzzy tolerance graphs by Pal et al. [22], fuzzy threshold graphs by Samanta et al. [28] etc. However, it has been observed that sometimes FGs become less effective and not applicable in dealing with several complex real-world problems. Consequently, numerous generalized forms of FGs have been explored in the literature. The very first generalization of FGs named interval-valued fuzzy graphs (IVFGs) was initiated by Akram et al. [1]. IVFGs is more effective and useful compared to the FGs, since in IVFGs we allocate subintervals of $[0, 1]$ instead of numbers. Hence the domain of IVFGs is more flexible and has more capacity to deal with vagueness. Afterward, many notions in the setting of IVFGs were initiated like highly irregular IVFGs by Rashmanlou et al. [21] and balanced IVFGs by Rashmanlou et al. [24]. IVFGs is also a one-sided case because it deals only with the degree of membership. In this era, we occasionally encounter more complex systems with more options and have observed that IVFGs are not effective in such scenarios. Hence further extension of FGs termed intuitionistic fuzzy graphs (IFGs) was explored by Parvathi et al. [19], it comprises degree of membership and degree of non-membership. IFGs allows us to deal with belonging (degree of membership) and non-belonging (degree of non-membership) instead of FGs which deals only with belonging (degree of membership). Further to this, several new notions of IFGs were introduced by Shao et al. [29], some operations on IFGs were defined by Thilagavathi et al. [30] and some operations on IFGs via novel versions of the Sombor index for internet routing were introduced by Imran et al. [13]. Subsequently, many terms of crisp graphs (CGs) were shifted in IFGs. The concept of covering in IFGs was initiated by Sahoo et al. [27]. Further to this, the extension of IFGs called interval-valued intuitionistic fuzzy graphs (IVIFGs) was introduced by Atanassov [4]. IVIFGs also deals with the degree of membership and degree of non-membership and allocates the values as sub-intervals of $[0, 1]$. But to make the IFGs more effective and usable, the term Pythagorean fuzzy graphs (PyFGs) was introduced by Verma et al. [31]. To extend the domain of PyFGs, the term IVPyFGs was proposed by Akram et al. [7]. An extension of both IFGs and PyFGs termed q-ROFGs was introduced by Habib et al. [12], it has more capacity to deal with real-world problems with uncertainties. Many researches have been conducted on q-ROFGs and its numerous applications have been explored like the application of q-ROFGs in DM was described by Atheeque et al. [5], q-ROFGs structures and their application towards DM were investigated by Akram et al. [6] etc. Recently, Jan et al. [16] initiated the concept of interval-valued q-rung orthopair fuzzy graphs (IVq-ROFGs). In IVq-ROFGs, the degree of membership and degree of non-membership are expressed in the form of sub-intervals of $[0, 1]$.

## 2 Literature review

FSs extends the concepts of classical sets to handle uncertainty and vagueness more precisely. FSs has proven a useful tool to solve the problems in various fields including artificial intelligence, control systems, decision-making, and data analysis. FSs assigns a degree of membership to each element in a given set $S$ from $[0, 1]$. Due to its flexible nature, numerous extensions of FSs like IVFSs, IFSs, IVIFSs etc have been explored. IVFSs enables the representation of uncertainty and imprecision in a more flexible manner. IVFSs utilizes degree of membership as a subinterval of $[0, 1]$ instead of a single value. Similarly, IFSs utilizes degree of membership and degree of non-membership as a single value from $[0, 1]$. IFSs provides us more comprehensive framework for handling vagueness. Furthermore, PyFSs is a cutting-edge extension of FSs which is more adaptable in handling uncertainties. PyFSs is based on the

notion of Pythagorean membership grades, where the degree of membership and degree of non-membership satisfy the condition $0 \leq (\omega)^2 + (\nu)^2 \leq 1$. Similarly, IVPyFSs is a significant addition in the theory of FSs, integrating the notion of IVFSs and PyFSs to effectively handle uncertainties. IVPyFSs assign interval-valued membership and non-membership degrees to each element. This innovative approach enables IVPyFSs to handle more complex sceneries that were not effectively handled by other generalizations of FSs like IFSs, PyFSs etc. Afterwards, the notion of q-ROFSs was established. q-ROFSs utilizes the degree of membership $(\omega)$ and degree of non-membership $(\nu)$ from $[0, 1]$ satisfying the condition $0 \leq (\omega)^q + (\nu)^q \leq 1$. A q-ROFSs is proven a cutting-edge extension of fuzzy set theory which provides a more flexible and effective framework for handling uncertainties. The model based on q-ROFSs has more capacity to deal such problems that were not dealt through traditional FSs like IFSs, PyFSs etc. Applications of q-ROFSs towards decision-making, image processing, data analysis, risk assessment, machine learning etc have been explored. By combining the concepts of IVFSs and q-ROFSs, the term IVq-ROFSs was introduced in the literature. IVq-ROFSs is the most flexible tool to deal with complex problems. IVq-ROFSs utilize degree of membership and degree of non-membership in the form of subintervals from $[0, 1]$. We can handle many real-world problems by using IVq-ROFSs more accurately. For more on IVq-ROFSs, one may consult [8].

FGs is the extension of classical graphs, incorporating the concepts of FSs to graphs. These models have more capacity to deal the problems containing uncertainties and vagueness. FGs assign fuzzy degrees of membership to vertices and edges, enabling the representation of imprecise relationships and uncertain connections. FGs has diverse applications including social network analysis, recommendation systems, image segmentation, data mining, network optimization etc. To make model of FGs more precise and flexible numerous terms have been introduced in the domain of FGs. One of the most important generalization of FGs named IVFGs was introduced by Akram et al. [1]. IVFGs assigns interval-valued membership degrees from $[0, 1]$ to vertices and edges, respectively. Many concepts of classical graphs have been studied in the domain of different FGs. The terms covering and matching are fundamental concepts in the theory of graphs that have been discussed in the paradigms of different extensions of FGs, and their applications including operations research, computer science and social network analysis have been explored.

Recently, the concepts of covering and matching in FGs were discussed in [18]. Further extending the notion of FGs, the notion of IFGs was introduced by Parvathi et al. [19], it utilized degree of membership and degree of non-membership for nodes and edges from $[0, 1]$. Several new terms including covering and matching IFGs were introduced in [27]. Further extended form of IFGs termed PyFGs was introduced by Verma et al. [31]. These concepts were also discussed for some other generalizations of FGs (see [15,23]). One of the most important extension of both IFGs and PyFGs named q-ROFGs was introduced by Habib et al. [12], it has more capacity to deal with real-world problems with uncertainties. Many studies have been conducted on q-ROFGs and its numerous applications have been explored [5,6]. Recently, Jan et al. [16] introduced the notion of IVq-ROFGs. IVq-ROFGs allocates subintervals of $[0, 1]$ as degree of membership and degree of non-membership for nodes and edges, respectively. Additionally, we elaborate the significance of study on IVq-ROFGs in Fig 1.

The abbreviations used in this manuscript are enlisted in Table 1.

**Motivations:** The concepts of covering and matching have been extensively discussed in classical graphs theory with many applications explored in different fields such as networking, data science, computer science etc have been explored. Recently, these concepts have been introduced in the paradigms of FGs, IFGs, PFGs, IVPFGs etc. The concepts of covering and

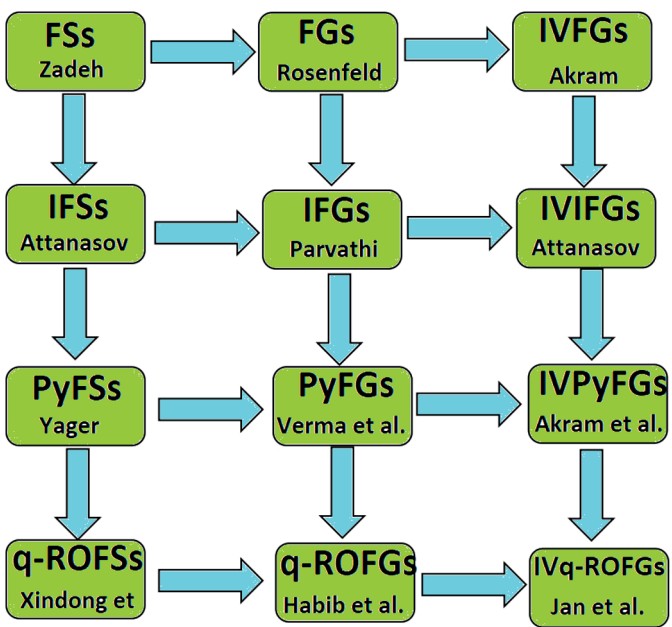

**Fig 1. Generalizations of the FSs and FGs.**

**Table 1. Abbreviations table.**

| Terminologies | Notations | Terminologies | Notations |
|---|---|---|---|
| Fuzzy sets | FSs | Pythagorean fuzzy graphs | PyFGs |
| Fuzzy graphs | FGs | Interval-valued pythagorean fuzzy graphs | IVPyFGs |
| Interval-valued fuzzy sets | IVFSs | q-rung orthopair fuzzy sets | q-ROFSs |
| Interval-valued fuzzy graphs | IVFGs | Interval-valued q-rung orthopair fuzzy sets | IVq-ROFSs |
| Intuitionistic fuzzy sets | IFSs | q-rung orthopair fuzzy graphs | q-ROFGs |
| Interval-valued intuitionistic fuzzy set | IVIFSs | Interval-valued q-rung orthopair fuzzy graphs | IVq-ROFGs |
| Imtuitionistic fuzzy graphs | IFGs | Strong arcs | SAs |
| interval-valued intuitionistic fuzzy graphs | IVIFGs | Strong arc cover | SAC |
| Pythagorean fuzzy sets | PyFSs | Strong node cover | SNC |
| Interval-valued pythagorean fuzzy sets | IVPyFSs | Strong matching | SM |
| Perfect strong matching | PSM | Strong independent set | SIS |
| Fuzzy air conditioning system | FACS | Interval-valued q-rung orthopair fuzzy weight | IVq-ROFW |
| Strongly independent | SI | Interval-valued q-rung orthopair picture fuzzy graphs | IVq-ROPFGs |
| Strong independent arc cover | SIAC | Max strong independent set | MSIS |
| Maximal strong independent set | MSIS | Fuzzy weight | FW |
| Membership function | MF | Non-membership function | NMF |

matching in FGs have played a vital role in modeling the real-world problems with uncertainties. On the other hand, IVq-ROFGs has a unique vast structure with the capability to handle uncertainties more accurately and slight changes in degree of membership allow it to be adapt

to other existing FGs models. Moreover, we observe that the notions of covering and matching have not yet been introduced in the domain of IVq-ROFGs. All of these factors motivated us to introduce these concepts in the realm of IVq-ROFGs and demonstrate their application.

**Novelty of our work:** We may describe the novelty of this work as follows.

1. We initiate the notions of SAC, SNC, SAC number, SIS and interval-valued q-rung orthopair fuzzy cycle (IVq-ROFC) by utilizing the SAs in the context of IVq-ROFGs and deduce many important results related to these terminologies.
2. The notions of complete IVq-ROFGs, complete bipartite IVq-ROFGs and cyclic IVq-ROFGs are introduced and investigated.
3. The concepts of matching, SM, PSM etc for IVq-ROFGs are explored.
4. To demonstrate the usefulness of the presented work, we provide an application towards the analysis of FACS with respect to the temperatures of different sensors. In this regard, we provide an algorithm which serves as a computational framework for FACS.
5. Finally, we conduct a comparative analysis of our presented model with the other existing models in the literature like IVIFGs, IVFGs and IVPyFGs.

**Organization of this work:** This manuscript is organized as: In Sect 2, we review the literature that is helpful for understanding the subsequent sections. Numerous useful terms related to FSs and FGs are presented in Sect 3. In Sect 4, we comprehensively discuss the notions of covering and matching in IVq-ROFGs. The terms like SNC, SAC and some useful notions like IVq-ROFC, SIS, SM and PSM based on SAs are also initiated. In Sect 5, we provide the application of covering in IVq-ROFGs towards the analysis of FACS. In Sect 6, we conduct a comparative analysis. We offer the conclusion of our study in Sect 7. We also provide the implications of our study including theoretical implications, practical implications, educational and industrial impacts etc. At the end, we provide a future work and directions of our study in Sect 8.

## 3 Preliminaries

In this section, we offer some helpful notions from the literature for further upcoming studies.

**Definition 1.** [34] A pair $(\omega, X)$ is a FSs on $X$, where $\omega : X \longrightarrow [0, 1]$ denotes the *MF*.

**Definition 2.** [35] An IVFSs $A$ defined on $U$ is expressed as $A = \{(m, [\omega_A^-(m), \omega_A^+(m)]) : m \in U\}$, where $\omega_A^- : X \to [0, 1]$ and $\omega_A^+ : X \to [0, 1]$ are $L$ and $U$ limits of degree of membership. Due to an interval $\omega_A^-(m) \leq \omega_A^+(m)$ for every $m \in U$.

**Definition 3.** [2] A set of the form of $\{(s, \omega_B(s), \nu_B(s) : s \in S)\}$ is called an IFSs, where $\omega_B(s) \in [0, 1]$ denotes the degree of membership of $s \in B$, $\nu_B(s) \in [0, 1]$ represents degree of non-membership of $s \in B$, with $0 \leq \omega_B(s) + \nu_B(s) \leq 1$, for all $s \in S$.

**Definition 4.** [3] An IVIFSs $S$ on $U$ is the set $\{(m), \omega_A(m), \nu_A(m)) : m \in M\}$, where
$\omega_A : U \to \mathrm{D}([0, 1]), \omega_A(w) = [\omega_A(w), \omega_A(w)] \in \mathrm{D}([0, 1])$
$\nu_A : U \to \mathrm{D}([0, 1]), \nu_A(w) = [\nu_A(w), \nu_A(w)] \in \mathrm{D}([0, 1])$ and
for all $w \in U, 0 \leq \omega_A(w) + \nu_A(w) \leq 1$.

**Definition 5.** [33] Let $\omega$ be the *MF* and $\nu$ be the *NMF* on universal set $U$. Then the PyFSs $S$ is $S = \{c, \omega_S(c), \nu_S(c) : c \in V\}$, where $\omega : V \to [0, 1]$ and $\nu : V \to [0, 1]$ such that $0 \leq (\omega_S(c))^2 + (\nu_S(c))^2 \leq 1$, for all $c \in V$.

**Definition 6.** [33] A PyFSs $R$ on $U \times U$ is called PyFR on $U$, described as $R = \{\langle xy, \omega_R(xy), \nu_R(xy)\rangle :$ $x, y \in U\}$, where $\omega_R : U \times U \to [0, 1]$ and $\nu_R : U \times U \to [0, 1]$ denote the degree of membership and degree of non-membership of $xy$ in $R$, respectively such that $0 \leq \omega_R^2(xy) + \nu_R^2(xy) \leq 1$, for all $x, y \in U$.

**Definition 7.** [20] A q-ROFR on set $U$ can be described as

$$R = [(\omega_R(x_i), \nu_R(x_i))|x_i \in U],$$

where $\omega_R$ and $\nu_R$ are the MF and NMF, respectively from the set $U \to [0, 1]$, where $0 \leq \omega_R^q(x_i) + \nu_R^q(x_i) \leq 1$. The term $\pi_R(x_i) = (1 - (\omega_R^q + \nu_R^q))^{1 \div q}$ is said to be an indeterminacy or hesitancy degree of element $x_i \in U$.

**Definition 8.** [11] An IVPyFSs on $U$ can be described as $S = \langle c, [\omega_S^L(c), \omega_S^U(c)], [\nu_S^L(c), \nu_S^U(c)]|c \in V\rangle$ with $0 \leq (\omega_R^U(c))^2 + (\nu_R^U(c))^2 \leq 1$.

**Definition 9.** [14] An IVq-ROFSs over $U$ is $A = \{x, (\omega(x), \nu(x))\}$ and $\omega$ and $\nu$ are mappings from $U \to D[0, 1]$, where $\omega(x) = [\omega^L(x), \omega^U(x)]$ and $\nu(x) = [\nu^L(x), \nu^U(x)]$ with $0 \leq (\omega^U(x))^q + (\nu^U(x))^q \leq 1$, for $q \in \mathbb{Z}^+$.

**Definition 10.** [25] A FGs $G = (M, N)$ with the functions $\nu : P \to [0, 1]$ and $\omega : P \times P \to [0, 1]$, where for $a, b$ in $P$, $\omega(a, b) \leq \nu(a) \wedge \nu(b)$.

**Definition 11.** [1] Let $G^\bullet = (V, E)$ be any graph. An IVFGs defined on $G^\bullet$ is $G = (M, N)$, where $M = [\omega_M^-, \omega_M^+]$ is an IVFSs on $V$ and $N = [\omega_N^-, \omega_M^+]$ is an IVFSs on $E$ such that $\omega_N^-(p, q) \leq \omega_M^-(p) \wedge \omega_M^-(q)$: $\omega_N^+(p, q) \leq \omega_M^+(p) \wedge \omega_M^+(q)$, for every $(p, q) \in E$.

**Definition 12.** [19] A graph of the form $G = (M, N)$ is an IFGs, where

1. $M = \{m_1, m_2, ..., m_n\}$ such that $\omega_1 : M \to [0, 1]$ and $\nu_1 : M \to [o, 1]$ denote the degree of membership and degree of non-membership of an element $m_i \in M$, respectively and

$$0 \leq \omega(m_i) + \nu(m_i) \leq 1 \tag{1}$$

   for all $m_i \in M$, (i= 1,2,3,...,n).
2. $N \in M \times M$ where $\omega_2 : M \times M \to [0, 1]$ and $\nu_2 : M \times M \to [0, 1]$ are such that

$$\omega_2(m_i, m_j) \leq \omega_1(m_i) \wedge \omega_1(m_j) \tag{2}$$
$$\nu_2(m_i, m_j) \leq \nu_1(m_i) \vee \nu_1(m_j) \tag{3}$$

   and

$$0 \leq \omega_2(m_i, m_j) + \nu_2(m_i, m_j) \leq 1 \tag{4}$$

   for every $(m_i, m_j) \in \mathbb{N}$, $(i, j = 1, 2, 3, ..., n)$.

**Definition 13.** [4] A pair $H = (M, N)$ is an IVIFGs defined on $G^\bullet = (V, E)$ such that $M = ([\omega_M^L, \omega_M^U], [\nu_M^L, \nu_M^U])$ is said to be an IVIFSs on $V$ and $N = ([\varepsilon_N^L, \varepsilon_N^U], [\nu_N^L, \nu_N^U])$ be an IVIFSs on $E \subseteq V \times V$, where $\omega_N^L(\grave{u}, \grave{v})$ and $\omega_N^U(\grave{u}, \grave{v})$ represent the lower degree of membership and

upper degree of membership and $\nu_N^L(\grave{u}, \grave{v})$ and $\nu_N^U(\grave{u}, \grave{v})$ represents the lower degree of non-membership and upper degree of non-membership, respectively such that for all edges $\grave{u}\grave{v} \in E$

$$\omega_N^L(\grave{u}, \grave{v}) \leq min(\omega_M^L(\grave{u}), \omega_M^L(\grave{v})), \omega_N^U(\grave{u}, \grave{v}) \leq min(\omega_M^U(\grave{u}), \omega_M^U(\grave{v}))$$
$$\nu_N^L(\grave{u}, \ \grave{v}) \geq max(\nu_M^L(\grave{u}), \nu_M^L(\grave{v})), \nu_N^U(\grave{u}, \ \grave{v}) \geq max(\nu_M^U(\grave{u}), \nu_M^U(\grave{v}))$$

satisfying $0 \leq \omega_N^U(\grave{u}, \grave{v}) + \nu_N^U(\grave{u}, \ \grave{v}) \leq 1$, for all $\grave{u}\grave{v} \in E$.

**Definition 14.**        [31] A PyFGs on $U$ is represented by $G = (V, S, R)$, where $S$ is a PyFSs on $U$ and $R$ be the PyFR on $U$ with

$\omega_R(xy) \leq \omega_S(c) \wedge \nu_S(c), \nu_R(xy) \leq \omega_S(c) \vee \nu_S(c)$ and $0 \leq \omega_R^2(xy) + \nu_R^2(xy) \leq 1$, also $\omega_R : U \times U \to [0, 1]$ and $\nu_R : U \times U \to [0, 1]$ denote degree of membership and degree of non-membership of $R$, respectively.

**Definition 15.**        [7] An IVPyFGs on $U$ is a pair $G = (S, R)$, where $S$ is the IVPyFSs and $R$ is a IVPyFR such that $\omega_R^L(xy) \leq \omega_S^L(x) \wedge \omega_S^L(y), \omega_R^U(xy) \leq \omega_S^U(x) \wedge \omega_S^U(y); \nu_R^L(xy) \geq \nu_S^L(x) \vee \nu_S^L(y), \nu_R^U(xy) \geq \nu_S^U(x) \vee \nu_S^U(y)$ and $0 \leq \omega_R^2(xy) + \nu_R^2(xy) \leq 1$, for all $x, y \in U$.

**Definition 16.** [12] A q-ROFGs on $U$ is a pair $G^\diamond = (G, F)$, where $G$ is a q-ROFSs on $U$ and $F$ is a q-ROFR on $U \times U$ such that

$$\omega_F^q(a, b) \leq \omega_G(a) \wedge \omega_G(b)$$
$$\nu_F^q(a, b) \geq \nu_G(a) \vee \nu_G(b)$$

and $0 \leq \omega_F^q(a, b) + \nu_F^q(a, b) \leq 1$, for all $a, b \in U$ and $q \geq 1$.

**Definition 17.** [16] A graph of the form $G^\diamond = (V^\diamond, E^\diamond)$ is said to be IVq-ROFGs, if

1. $V^\diamond = f_1, f_1, d_3, ..., f_n$ such that $\omega_{PL} : V^\diamond \to D[0, 1], \omega_{PU} : V^\diamond \to D[0, 1]$ represent the lower degree of membership and upper degree of membership and $\nu_{PL} : V^\diamond \to D[0, 1], \nu_{PU} : V^\diamond \to D[0, 1]$ denote the lower degree of non-membership and upper degree of non-membership of $f_i \in V^\diamond$, respectively where $0 \leq (\omega_{PU})^q + (\nu_{PU})^q \leq 1$, for $q \in \mathbb{Z}^+$, for all $f_i \in V^\diamond, (i = 1, 2, 3, ..., m)$

2. $E^\diamond \subseteq V^\diamond \times V^\diamond$, where $\omega_{QL}, \nu_{QL} : V^\diamond \times V^\diamond \to D[0, 1]$ and $\omega_{QU}, \nu_{QU} : V^\diamond \times V^\diamond \to D[0, 1]$ such that $\omega_{QL}(f_i, f_j) \leq \omega_{PL}(f_i) \wedge \omega_{PL}(f_j), \omega_{QU}(f_i, f_j) \leq \omega_{PU}(f_i) \wedge \omega_{PU}(f_j)$ where $\omega_{QU}(f_i, f_j) \geq \omega_{PL}(f_i) \wedge \omega_{PL}(f_j)$ and $\nu_{QL}(f_i, f_j) \leq \nu_{PL}(f_i) \vee \nu_{PL}(f_j), \nu_{QU}(f_i, f_j) \leq \nu_{PU}(f_i) \vee \nu_{PU}(f_j)$ such that $\nu_{QU}(f_i, f_j) \geq \nu_{PL}(f_i) \vee \nu_{PL}(f_j)$ with $0 \leq (\omega_{QU}(f_i, f_j))^q + (\nu_{QU}(f_i, f_j))^q \leq 1, q \in \mathbb{Z}^+$, for all $(f_i, f_j) \in E^\diamond$.

## 4 Covering and matching in IVq-ROFGs

In this portion, we introduce some useful definitions, results and the notions related to covering and matching in IVq-ROFGs.

**Definition 18.** The SAs of an IVq-ROFGs is an edge $(f_i, f_j)$ with

$$\omega_Q^L(f_i, f_j) = min(\omega_P^L(f_i), \omega_P^L(f_j)), \omega_Q^U(f_i, f_j) = min(\omega_P^U(f_i), \omega_P^U(f_j))$$
$$\nu_Q^L(f_i, f_j) = max(\nu_P^L(f_i), \nu_P^L(f_j)), \nu_Q^U(f_i, f_j) = max(\nu_P^U(f_i), \nu_P^U(f_j)).$$

where $\omega_Q^L$ and $\nu_Q^L$ are the lower degree of membership and lower degree of non-membership, respectively.

Now we introduce the notion of SNC in IVq-ROFGs with the help of SAs.

**Definition 19.** Let $G^\circ = (V^\diamond, E^\diamond)$ be an IVq-ROFGs. In an IVq-ROFGs $G^\circ$, the collection of vertices $Z$ that encompasses all SAs of $G^\circ$ is called SNC. An IVq-ROFW $\check{W}$ of SNC $Z$ is describes as

$$\check{W}_{nc} = \left\langle \left[ \check{W}_{nc}^{L\omega}(Z), \check{W}_{nc}^{U\omega}(Z) \right], \left[ \check{W}_{nc}^{L\nu}(Z), \check{W}_{nc}^{U\nu}(Z) \right] \right\rangle$$

$$\check{W}_{nc} = \left\langle \left[ \sum_{f_i \in Z} \omega_{E^\diamond}^L(f_i, f_j), \sum_{f_i \in Z} \omega_{E^\diamond}^U(f_i, f_j) \right], \left[ \sum_{f_i \in Z} \nu_{E^\diamond}^L(f_i, f_j), \sum_{f_i \in Z} \nu_{E^\diamond}^U(f_i, f_j) \right] \right\rangle$$

where $\omega_{E^\diamond}^L(f_i, f_j)$ and $\omega_{E^\diamond}^U(f_i, f_j)$ denote the min of the lower degree of membership and upper degree of membership of all SAs and $\nu_{E^\diamond}^L(f_i, f_j)$, $\nu_{E^\diamond}^U(f_i, f_j)$ represent the max of the lower degree of non–membership and upper degree of non–membership of all of SAs in an IVq–ROFGs $G^\circ$ that are incident to $f_i$.

**Definition 20.** A set $\lambda_0(G^\circ) = \lambda_0 = \langle [\lambda_{10}^{L\omega}, \lambda_{10}^{U\omega}], [\lambda_{20}^{L\nu}, \lambda_{20}^{U\nu}] \rangle$ is a SNC number of IVq-ROFGs, if

$$\lambda_{10}^{L\omega} = min\{ \check{W}_{nc}^{L\omega}(Z) | Z \text{ is the SNC of } G^\circ \}$$

$$\lambda_{10}^{U\omega} = min\{ \check{W}_{nc}^{U\omega}(Z) | Z \text{ is the SNC of } G^\circ \}$$

$$\lambda_{20}^{L\nu} = max\{ \check{W}_{nc}^{L\nu}(Z) | Z \text{ is the SNC of } G^\circ \}$$

$$\lambda_{20}^{U\nu} = max\{ \check{W}_{nc}^{U\nu}(Z) | Z \text{ is the SNC of } G^\circ \}$$

where FW is the notation of fuzzy weight and the min degree of membership and max degree of non-membership of SNC is called a min SNC in IVq-ROFGs $G^\circ$.

**Definition 21.** An IVq-ROFGs is a complete IVq-ROFGs, if

$$\omega_{E^\diamond}^L(f_i, f_j) = min(\omega_{V^\diamond}^L(f_i), \omega_{V^\diamond}^L(f_j)), \omega_{E^\diamond}^U(f_i, f_j) = min(\omega_{V^\diamond}^U(f_i), \omega_{V^\diamond}^U(f_j))$$
$$\nu_{E^\diamond}^L(f_i, f_j) = max(\nu_{V^\diamond}^L(f_i), \nu_{V^\diamond}^L(f_j)), \nu_{E^\diamond}^U(f_i, f_j) = max(\nu_{V^\diamond}^U(f_i), \nu_{V^\diamond}^U(f_j))$$

for all $(f_i, f_j) \in E^\diamond$ where

$$\omega_{QU}(f_i, f_j) > \omega_{PL}(f_i) \wedge \omega_{PL}(f_j)$$
$$\nu_{QU}(f_i, f_j) > \nu_{PL}(f_i) \vee \nu_{PL}(f_j)$$

satisfying $0 \leq (\omega_{QU}(f_i, f_j))^q + (\nu_{QU}(f_i, f_j))^q \leq 1$.

**Definition 22.** An IVq-ROFGs that can be partitioned into subsets $V_1^\diamond$ and $V_2^\diamond |$ for $f_i f_j \in V_1^\diamond$ or $f_i f_j \in V_2^\diamond$, $\omega_{E^\diamond}^L(f_i, f_j) = \omega_{E^\diamond}^U(f_i, f_j) = 0$, and $\nu_{E^\diamond}^L(f_i, f_j) = \nu_{E^\diamond}^U(f_i, f_j) = 0$, is said to be a bipartite IVq-ROFGs, if for all $f_i \in V_1^\diamond$ and $f_j \in V_2^\diamond$,

$$\omega_{E^\diamond}^L(f_i, f_j) = min(\omega_{V^\diamond}^L(f_i), \omega_{V^\diamond}^L(f_j)), \omega_{E^\diamond}^U(f_i, f_j) = min(\omega_{V^\diamond}^U(f_i), \omega_{V^\diamond}^U(f_j))$$

$$\nu^L_{E^\diamond}(f_i, f_j) = max(\nu^L_{V^\diamond}(f_i), \nu^L_{V^\diamond}(f_j)), \ \nu^U_{E^\diamond}(f_i, f_j) = max(\nu^U_{V^\diamond}(f_i), \nu^U_{V^\diamond}(f_j))$$

then $G^\circ$ is called a complete bipartite IVq-ROFGs.

**Definition 23.** An edge (arc) of the form $(f_i, f_j)$ with $\omega^{LQ}_{UQ}(f_i, f_j), \nu^{LQ}_{UQ}(f_i, f_j) > 0$ is said to be a weakest arc of IVq–ROFGs $G^\circ$ if is an edge with min $\omega^{LQ}_{UQ}(f_i, f_j)$ and min $\nu^{LQ}_{UQ}(f_i, f_j)$.

**Definition 24.** A path S of lengths is a sequence of vertices $f_1, f_2, f_3, ..., f_s | \omega^{LQ}_{UQ}(f_{j-1}, f_j), \nu^{LQ}_{UQ}(f_{j-1}, f_j) > 0, j = 1, 2, 3, ..., s$ and the lower and upper degree of membership and lower and upper degree of non-membership of weakest arc is defined as its strength. If $f_0 = f_s$ and $s \geq 3$ then S is said to be a cycle and $\dot{G}$ is called interval-valued q-rung orthopair fuzzy cycle (IVq-ROFC), if it contains more then one weakest edge (arc). The strength of a cycle is the strength of a weakest edge in it.

**Theorem 25.** *Let $G^\circ = (V^\diamond, E^\diamond)$ be a complete IVq–ROFGs with number of nodes s. Then*

$$\lambda^{L\omega}_{10} = (s-1)\omega^L_{E^\diamond}(f_i, f_j)$$
$$\lambda^{L\nu}_{20} = (s-1)\nu^L_{E^\diamond}(f_i, f_j)$$
$$\lambda^{U\omega}_{10} = (s-1)\omega^U_{E^\diamond}(f_i, f_j)$$
$$\lambda^{U\nu}_{20} = (s-1)\nu^U_{E^\diamond}(f_i, f_j)$$

*where $\omega^L_{E^\diamond}(f_i, f_j))$ and $\omega^U_{E^\diamond}(f_i, f_j))$ denote lower degree of membership and upper degree of membership and $\nu^L_{E^\diamond}(f_i, f_j)), \nu^U_{E^\diamond}(f_i, f_j))$ express the lower degree of non–membership and upper degree of non–membership of the least SAs in $G^\circ$.*

*Proof*: Let $G^\circ$ be a complete IVq-ROFGs. Then it is clear that each vertex associated to $G^\circ$ are associated through SAs. Thus either the collection $(s-1)$ of vertices is SNC of $G^\circ$. Suppose a vertex in $G^\circ$ with min degree of membership and max degree of non–membership is $f_i$ such that $f_i$ is connected to the distinct vertices $f_{j1}, f_{j2}, f_{j3}..., f_{js-1}$. Then from all the least SAs of $G^\circ$, $(f_i, f_{j1}), (f_i, f_{j2}), ..., (f_i, f_{js-1})$ is the arc having degree of membership $[\omega^L_{E^\diamond}(f_i, f_j), \omega^U_{E^\diamond}(f_i, f_j)]$ and degree of non–membership $[\nu^L_{E^\diamond}(f_i, f_j), \nu^U_{E^\diamond}(f_i, f_j)]$, where $f_j \in f_{j1}, f_{j2}, f_{j3}..., f_{js-1}$. Hence the set $Z$ of vertices is the set $f_{j1}, f_{j2}, f_{j3}..., f_{js-1}$ forms SNC of $G^\circ$ with

$$\check{W}^{L\omega}_{nc}(Z) = \sum_{f_j \in Z} \omega^L_{E^\diamond}(f_i, f_j) = \omega^L_{E^\diamond}(f_i, f_{j1}) + \omega^L_{E^\diamond}(f_i, f_{j2}) + ... + \omega^L_{E^\diamond}(f_i, f_{js-1})$$
$$\check{W}^{U\omega}_{nc}(Z) = \sum_{f_j \in Z} \omega^U_{E^\diamond}(f_i, f_j) = \omega^U_{E^\diamond}(f_i, f_{j1}) + \omega^U_{E^\diamond}(f_i, f_{j2}) + ... + \omega^U_{E^\diamond}(f_i, f_{js-1})$$

where $\omega^L_{E^\diamond}(f_i, f_j), j = 1, 2, ..., (s-1)$ and $\omega^U_{E^\diamond}(f_i, f_j), j = 1, 2, ..., (s-1)$ denote the min of the lower degree of membership and upper degree of membership of SAs incident on $f_j$, respectively. Then

$$\lambda^{L\omega}_{10} = \omega^L_{E^\diamond}(f_i, f_j) + \omega^L_{E^\diamond}(f_i, f_j) + \omega^L_{E^\diamond}(f_i, f_j)$$
$$\lambda^{U\omega}_{10} = \omega^U_{E^\diamond}(f_i, f_j) + \omega^U_{E^\diamond}(f_i, f_j) + \omega^U_{E^\diamond}(f_i, f_j)$$

where $\omega^L_{E^\diamond}(f_i, f_j)$ and $\omega^U_{E^\diamond}(f_i, f_j)$ denote the lower degree of membership and upper degree of membership of least SAs in IVq–ROFGs $G^\circ$. Hence

$$\lambda^{L\omega}_{10} = (s-1)\omega^L_{E^\diamond}(f_i, f_j)$$
$$\lambda^{U\omega}_{10} = (s-1)\omega^U_{E^\diamond}(f_i, f_j)$$

Similarly

$$\check{W}_{nc}^{L\nu}(Z) = \sum_{f_j \in Z} \nu_{E\diamond}^L(f_i, f_j) = \nu_{E\diamond}^L(f_i, f_{j1}) + \nu_{E\diamond}^L(f_i, f_{j2}) + ... + \nu_{E\diamond}^L(f_i, f_{js-1})$$

$$\check{W}_{nc}^{U\nu}(Z) = \sum_{f_j \in Z} \nu_{E\diamond}^U(f_i, f_j) = \nu_{E\diamond}^U(f_i, f_{j1}) + \nu_{E\diamond}^U(f_i, f_{j2}) + ... + \nu_{E\diamond}^U(f_i, f_{js-1})$$

where $\nu_{E\diamond}^L(f_i, f_j), j = 1, 2, ..., (s-1)$ and $\nu_{E\diamond}^U(f_i, f_j), j = 1, 2, ..., (s-1)$ denote the max lower degree of non–membership and upper degree of non–membership of the SAs incident on $f_j$, correspondingly. Then

$$\lambda_{20}^{L\nu} = \nu_{E\diamond}^L(f_i, f_j) + \nu_{E\diamond}^L(f_i, f_j) + \nu_{E\diamond}^L(f_i, f_j)$$

$$\lambda_{20}^{U\nu} = \nu_{E\diamond}^U(f_i, f_j) + \nu_{E\diamond}^U(f_i, f_j) + \nu_{E\diamond}^U(f_i, f_j)$$

where $\nu_{E\diamond}^L(f_i, f_j)$ and $\nu_{E\diamond}^U(f_i, f_j)$ denote the lower degree of non–membership and upper degree of non–membership of least SAs in IVq–ROFGs $G^\circ$. Hence

$$\lambda_{20}^{L\nu} = (s-1)\nu_{E\diamond}^L(f_i, f_j)$$

$$\lambda_{20}^{U\nu} = (s-1)\nu_{E\diamond}^U(f_i, f_j).$$

□

**Theorem 26.** *If a complete bipartite IVq-ROFGs $\check{G}$ is divided into two subsets $V_1^\circ$ and $V_2^\circ$, then*

$$\lambda_{10}^{L\omega}(\check{G}) = min\{\check{W}_{nc}^{L\omega}(V_1^\circ), \check{W}_{nc}^{L\omega}(V_2^\circ)\}$$

$$\lambda_{10}^{U\omega}(\check{G}) = min\{\check{W}_{nc}^{U\omega}(V_1^\circ), \dot{W}_{nc}^{U\omega}(V_2^\circ)\}$$

$$\lambda_{20}^{L\nu}(\check{G}) = max\{\check{W}_{nc}^{L\nu}(V_1^\circ), \check{W}_{nc}^{L\nu}(V_2^\circ)\}$$

$$\lambda_{20}^{U\nu}(\check{G}) = max\{\check{W}_{nc}^{U\nu}(V_1^\circ), \check{W}_{nc}^{U\nu}(V_2^\circ)\}$$

*Proof*: Let $\check{G}$ be a complete bipartite IVq-ROFGs, then clearly all of the arcs of $\check{G}$ are SAs. Assume that $V_1^\circ$ and $V_2^\circ$ are two subsets of the set of vertices of $\check{G}$ such that every vertex in $V_1^\circ$ is connected to all vertices in $V_2^\circ$. The collection of arcs of a complete bipartite graph $\check{G}$ is the union of the collection of all arcs incident to each vertex of $V_1^\circ$ and the collection of all arcs incident to each vertex of $V_2^\circ$. Moreover, $V_1^\circ$, $V_2^\circ$ and their union $V_1^\circ \cup V_2^\circ$ are IVq–ROFGs in $\check{G}$. Obviously,

$$\check{W}_{nc}^{L\omega}(V_1^\circ \cup V_2^\circ) > \check{W}_{nc}^{L\omega}(V_1^\circ)$$

$$\check{W}_{nc}^{L\omega}(V_1^\circ \cup V_2^\circ) > \check{W}_{nc}^{L\omega}(V_2^\circ)$$

Hence

$$\lambda_{10}^{L\omega}(\check{G}) = min\{\check{W}_{nc}^{L\omega}(V_1^\circ), \check{W}_{nc}^{L\omega}(V_2^\circ)\}.$$

Similarly,

$$\lambda_{10}^{U\omega}(\check{G}) = min\{\check{W}_{nc}^{U\omega}(V_1^\circ), \check{W}_{nc}^{U\omega}(V_2^\circ)\}$$

Also, we have

$$\check{W}_{nc}^{L\nu}(V_1^\circ \cup V_2^\circ) < \check{W}_{nc}^{L\nu}(V_1^\circ)$$

$$\check{W}_{nc}^{L\nu}(V_1^\circ \cup V_2^\circ) < \check{W}_{nc}^{L\nu}(V_2^\circ)$$

Hence

$$\lambda_{20}^{L\nu}(\check{G}) = max\{\check{W}_{nc}^{L\nu}(V_1^\diamond), \check{W}_{nc}^{L\nu}(V_2^\diamond)\}$$

Similarly, we have

$$\lambda_{20}^{U\nu}(\check{G}) = max\{\check{W}_{nc}^{U\nu}(V_1^\diamond), \check{W}_{nc}^{U\nu}(V_2^\diamond)\}.$$

□

**Proposition 27.** *If $\dot{G}$ be an IVq–ROFC such that $\vec{G}$ be a cycle with $\hat{n}$ number of vertices in $\dot{G}$, then*

$$\lambda_{11}^{L\omega} = min\{\check{W}_{nc}^{L\omega}(Z)|Z \text{ is SNC of } \dot{G} \text{ with } |Z| \geq \lceil \tfrac{\hat{n}}{2} \rceil\}$$

$$\lambda_{11}^{U\omega} = min\{\check{W}_{nc}^{U\omega}(Z)|Z \text{ is SNC of } \dot{G} \text{ with } |Z| \geq \lceil \tfrac{\hat{n}}{2} \rceil\}$$

$$\lambda_{21}^{L\nu} = max\{\check{W}_{nc}^{L\nu}(Z)|Z \text{ is SNC of } \dot{G} \text{ with } |Z| \geq \lceil \tfrac{\hat{n}}{2} \rceil\}$$

$$\lambda_{21}^{U\nu} = max\{\check{W}_{nc}^{U\nu}(Z)|Z \text{ is SNC of } \dot{G} \text{ with } |Z| \geq \lceil \tfrac{\hat{n}}{2} \rceil\}$$

*Proof*: In an IVq-ROFC $\dot{G}$, all of the arcs are SAs. Due to this, the no. of vertices in SNC are same in $\dot{G}$ and $\vec{G}$. Now, the SNC number of $\vec{G}$ is $\lceil \tfrac{\hat{n}}{2} \rceil\}$. Thus, the min no. of vertices in SNC of $\dot{G}$ is $\lceil \tfrac{\hat{n}}{2} \rceil\}$.                                    □

**Definition 28.**     Let $G^\circ$ be an IVq-ROFGs. Then two nodes in $G^\circ$ are called SI, if there no SAs between these two nodes. If the collection of vertices have two vertices that are SI, then the collection (set) is called a SIS.

**Definition 29.**  An IVq-ROFW of SIS $Z$ in an IVq–ROFGs $G^\circ$ can be described as

$$\check{W}_{is} = \left\langle \left[\check{W}_{is}^{L\omega}(Z), \check{W}_{is}^{U\omega}(Z)\right], \left[\check{W}_{is}^{L\nu}(Z), \check{W}_{is}^{U\nu}(Z)\right]\right\rangle$$

$$\check{W}_{is} = \left\langle \left[\sum_{f_i \in Z} \omega_{E^\diamond}^L(f_i, f_j), \sum_{f_i \in Z} \omega_{E^\diamond}^U(f_i, f_j)\right], \left[\sum_{f_i \in Z} \nu_{E^\diamond}^L(f_i, f_j), \sum_{f_i \in Z} \nu_{E^\diamond}^U(f_i, f_j)\right]\right\rangle$$

where $\omega_{E^\diamond}^L(f_i, f_j)$ and $\omega_{E^\diamond}^U(f_i, f_j)$ denote the min of the lower degree of membership and upper degree of membership of the SAs and $\nu_{E^\diamond}^L(f_i, f_j)$ and $\eta_{E^\diamond}^U(f_i, f_j)$ represent the max of the lower degree of non–membership and upper degree of non–membership of the SAs of IVq–ROFGs $G^\circ$, which are incident on $f_i$.

SI number of IVq-ROFGs is represented by $\varsigma_0(G^\circ) = \varsigma_0 = \langle [\varsigma_{10}^{L\omega}, \varsigma_{10}^{U\omega}], [\varsigma_{20}^{L\nu}, \varsigma_{20}^{U\nu}]\rangle$ with

$$\varsigma_{10}^{L\omega} = max\{\check{W}_{is}^{L\omega}(Z)|Z \text{ is the SIS of vertices in } G^\circ\}$$

$$\varsigma_{10}^{U\omega} = max\{\check{W}_{is}^{U\omega}(Z)|Z \text{ is the SIS of vertices in } G^\circ\}$$

$$\varsigma_{20}^{L\nu} = min\{\check{W}_{is}^{L\nu}(Z)|Z \text{ is the SIS of vertices in } G^\circ\}$$

$$\varsigma_{20}^{U\nu} = min\{\check{W}_{is}^{U\nu}(Z)|Z \text{ is the SIS of vertices in } G^\circ\}$$

SIS having max degree of membership and min degree of non-membership is said to be an MSIS in IVq-ROFGs $G^\circ$.

**Proposition 30.** *If $\dot{G}$ be an IVq–ROFC such that $\vec{G}$ be a cycle with $\hat{n}$ number of vertices in $\dot{G}$, then*

$$\varsigma_{10}^{L\omega} = max\{\check{W}_{is}^{L\omega}(Z)|Z \text{ is the SIS of vertices in } \dot{G} \text{ with } |Z| \leq \lceil \tfrac{\check{n}}{2} \rceil\}$$

$$\varsigma_{10}^{U\omega} = max\{\check{W}_{is}^{U\omega}(Z)|Z \text{ is the SIS of vertices in } \dot{G} \text{ with } |Z| \leq \lceil \tfrac{\check{n}}{2} \rceil\}$$

$$\varsigma_{20}^{L\nu} = min\{\check{W}_{is}^{L\nu}(Z)|Z \text{ is the SIS of vertices in } \dot{G} \text{ with } |Z| \leq \lceil \tfrac{\check{n}}{2} \rceil\}$$

$$\varsigma_{20}^{U\nu} = min\{\check{W}_{is}^{U\nu}(Z)|Z \text{ is the SIS of vertices in } \dot{G} \text{ with } |Z| \leq \lceil \tfrac{\check{n}}{2} \rceil\}$$

*Proof*: In an IVq-ROFC $\dot{G}$, all of the arcs are SAs. Due to this, the no. of vertices in SIS of both $\dot{G}$ and $\vec{G}$ are same. Now, the SIS number of $\vec{G}$ is $\lceil \tfrac{\check{n}}{2} \rceil\}$. Thus, the max no. of vertices in SIS of $\dot{G}$ is $\lceil \tfrac{\check{n}}{2} \rceil\}$. □

**Theorem 31.** *Let* $G° = (V^\diamond, E^\diamond)$ *be a complete IVq-ROFGs. Then*

$$\varsigma_0(G°) = \langle[\omega_{E^\diamond}^L(f_i, f_j), \omega_{E^\diamond}^U(f_i, f_j)], [\nu_{E^\diamond}^L(f_i, f_j), \nu_{E^\diamond}^U(f_i, f_j)]\rangle$$

*where* $\omega_{E^\diamond}^L(f_i, f_j)$ *and* $\omega_{E^\diamond}^U(f_i, f_j)$ *denote the lower degree of membership and upper degree of membership and* $\nu_{E^\diamond}^L(f_i, f_j)$ *and* $\nu_{E^\diamond}^U(f_i, f_j)$ *represent the lower degree of non–membership and upper degree of non–membership of the least SAs in* $G°$, *respectively.*

*Proof*: Assume that $G°$ is a complete IVq-ROFGs. Then each of its arc is SAs and all of the vertices are linked to all other vertices in $G°$. Hence, $Z = \{f_j\}$ *is the only SIS for every* $f_j \in V^\diamond$. □

**Theorem 32.** *If a complete bipartite IVq-ROFGs* $\check{G}$ *is divided into two subsets* $V_1^\diamond$ *and* $V_2^\diamond$, then

$$\varsigma_{10}^{L\omega}(\check{G}) = min\{\check{W}_{is}^{L\omega}(V_1^\diamond), \check{W}_{is}^{L\omega}(V_2^\diamond)\}$$

$$\varsigma_{10}^{U\omega}(\check{G}) = min\{\check{W}_{is}^{U\omega}(V_1^\diamond), \check{W}_{is}^{U\omega}(V_2^\diamond)\}$$

$$\varsigma_{20}^{L\nu}(\check{G}) = max\{\check{W}_{is}^{L\nu}(V_1^\diamond), \check{W}_{is}^{L\nu}(V_2^\diamond)\}$$

$$\varsigma_{20}^{U\nu}(\check{G}) = max\{\check{W}_{is}^{U\nu}(V_1^\diamond), \check{W}_{is}^{U\nu}(V_2^\diamond)\}$$

*Proof*: Let $\check{G}$ be a complete bipartite IVq-ROFGs. Then every arc in $\check{G}$ is SA. Since each of the node in $V_1^\diamond$ is linked to all other nodes in $V_1^\diamond$, and each of the vertex in $V_2^\diamond$ is linked to every other vertex in $V_1^\diamond$. Therefore, $V_1^\diamond$ and $V_2^\diamond$ are said to be the SIS in $\check{G}$. □

**Theorem 33.** *Let* $G°$ *be an IVq-ROFGs without any isolated vertex. Then*

$$\lambda_{10}^{L\omega} + \varsigma_{10}^{L\omega} = \check{W}^{L\omega}(V^\diamond)$$

$$\lambda_{20}^{L\nu} + \varsigma_{20}^{L\nu} = \check{W}^{L\nu}(V^\diamond)$$

$$\lambda_{10}^{U\omega} + \varsigma_{10}^{L\omega} = \check{W}^{U\omega}(V^\diamond)$$

$$\lambda_{20}^{U\nu} + \varsigma_{20}^{L\nu} = \check{W}^{U\nu}(V^\diamond)$$

*Proof*: Let min SNC of $G°$ is $N_0$. Then

$$\lambda_{10}^{L\omega} = \check{W}^{L\omega}(N_0)$$

$$\lambda_{20}^{L\nu} = \check{W}^{L\nu}(N_0)$$

$$\lambda_{10}^{U\omega} = \check{W}^{U\omega}(N_0)$$

$$\lambda_{20}^{U\nu} = \check{W}^{U\nu}(N_0)$$

and $V^\diamond - N_0$ forms SIS of vertices. Moreover, $V^\diamond - N_0$ contains the vertices which are incident on SAs of $G^\circ$. Thus

$$
\begin{aligned}
\varsigma_{10}^{L\omega} &\geq \check{W}^{L\omega}(V^\diamond - N_0) = \check{W}^{L\omega}(V^\diamond) - \lambda_{10}^{L\omega} \Rightarrow \lambda_{10}^{L\omega} + \varsigma_{10}^{L\omega} \geq \check{W}^{L\omega}(V^\diamond) \\
\varsigma_{10}^{U\omega} &\geq \check{W}^{U\omega}(V^\diamond - N_0) = \check{W}^{U\omega}(V^\diamond) - \lambda_{10}^{U\omega} \Rightarrow \lambda_{10}^{U\omega} + \varsigma_{10}^{U\omega} \geq \check{W}^{U\omega}(V^\diamond) \\
\varsigma_{20}^{L\nu} &\leq \check{W}^{L\nu}(V^\diamond - N_0) = \check{W}^{L\nu}(V^\diamond) - \lambda_{20}^{L\nu} \Rightarrow \lambda_{20}^{L\nu} + \varsigma_{20}^{L\nu} \leq \check{W}^{L\nu}(V^\diamond) \\
\varsigma_{20}^{U\nu} &\leq \check{W}^{L\nu}(V^\diamond - N_0) = \check{W}^{U\nu}(V^\diamond) - \lambda_{20}^{U\nu} \Rightarrow \lambda_{20}^{U\nu} + \varsigma_{20}^{U\nu} \leq \check{W}^{U\nu}(V^\diamond)
\end{aligned}
\tag{5}
$$

Let MSIS of $G^\circ$ is $P_0$ i.e., $P_0$ having the vertices which are not neighbor based on SAs. Thus $V^\diamond - P_0$ having the vertices which cover all of the SAs of $G^\diamond$. Hence, $V^\diamond - P_0$ is an SNC of $G^\circ$, where $\lambda_{10}^{L\omega}$ and $\lambda_{10}^{U\omega}$ are the min lower degree of membership and upper degree of membership and $\lambda_{20}^{L\nu}, \lambda_{20}^{U\nu}$ are the max lower degree of non-membership and upper degree of non-membership, respectively. Then

$$
\begin{aligned}
\lambda_{10}^{L\omega} &\leq \check{W}^{L\omega}(V^\diamond - P_0) = \check{W}^{L\omega}(V^\diamond) - \varsigma_{10}^{L\omega} \Rightarrow \lambda_{10}^{L\omega} + \varsigma_{10}^{L\omega} \leq \check{W}^{L\omega}(V^\diamond) \\
\lambda_{10}^{U\omega} &\leq \check{W}^{U\omega}(V^\diamond - P_0) = \check{W}^{U\omega}(V^\diamond) - \varsigma_{10}^{U\omega} \Rightarrow \lambda_{10}^{U\omega} + \varsigma_{10}^{U\omega} \leq \check{W}^{U\omega}(V^\diamond) \\
\lambda_{20}^{L\nu} &\geq \check{W}^{L\nu}(V^\diamond - P_0) = \check{W}^{L\nu}(V^\diamond) - \varsigma_{20}^{L\nu} \Rightarrow \lambda_{20}^{L\nu} + \varsigma_{20}^{L\nu} \geq \check{W}^{L\nu}(V^\diamond) \\
\lambda_{20}^{U\nu} &\geq \check{W}^{L\nu}(V^\diamond - P_0) = \check{W}^{U\nu}(V^\diamond) - \varsigma_{20}^{U\nu} \Rightarrow \lambda_{20}^{U\nu} + \varsigma_{20}^{U\nu} \geq \check{W}^{U\nu}(V^\diamond)
\end{aligned}
\tag{6}
$$

From (5) and (6), we have

$$
\begin{aligned}
\lambda_{10}^{L\omega} + \varsigma_{10}^{L\omega} &= \check{W}^{L\omega}(V^\diamond), \quad \lambda_{10}^{U\omega} + \varsigma_{10}^{U\omega} = \check{W}^{U\omega}(V^\diamond) \\
\lambda_{20}^{L\nu} + \varsigma_{20}^{L\nu} &= \check{W}^{L\nu}(V^\diamond), \quad \lambda_{20}^{U\nu} + \varsigma_{20}^{U\nu} = \check{W}^{U\nu}(V^\diamond)
\end{aligned}
$$

Hence the proof is completed.                                                                 □

**Example 34.**      Consider an IVq-ROFGs provided in Fig 2. All of its arcs are SAs and all the SNC of IVq-ROFGs $G^\circ$ are: $Z_1 = \{a, c\}$, $Z_2 = \{a, b, c\}$, $Z_3 = \{c, d, a\}$, $Z_4 = \{a, c, d\}$, $Z_5 = \{b, a, c\}$, $Z_6 = \{a, b, c, d\}$. Table 2 and Table 3 denote the calculations regarding IVq-ROFW of SNC number $\check{W}_{nc}(Z)$ of an IVq - ROFGs $G^\circ$. Hence $\lambda_0(G^\circ) = \langle [0.2, 1.0], [2.6, 3.2] \rangle$.

**Example 35.**  Let us consider an IVq-ROFGs is strong demonstrated in Fig 3. The collection of SISs are: $Z_1 = \{b, d\}$, $Z_2 = \{a, c\}$ calculations regarding the FW of SI number of IVq - ROFGs $G^\circ$ are depicted in Table 4. Hence $\varsigma_0(G^\circ) = \langle [0.5, 1.4], [1.0, 1.8] \rangle$.

**Definition 36.**      Let $G^\circ$ be a IVq-ROFGs without any isolated vertex. In IVq-ROFGs $G^\circ$, an SAC is the set of SAs that covers all vertices of IVq-ROFGs $G^\circ$. The FW $\check{W}$ of SAC $\hat{A}$ is described as

$$
\check{W}_{ac} = \left\langle \left[ \check{W}_{ac}^{L\omega}(\hat{A}), \check{W}_{ac}^{U\omega}(\hat{A}) \right], \left[ \check{W}_{ac}^{L\nu}(\hat{A}), \check{W}_{ac}^{U\nu}(\hat{A}) \right] \right\rangle
$$

$$
\check{W}_{ac} = \left\langle \left[ \sum_{(f_i, f_j) \in \hat{A}} \omega_{E^\diamond}^L(f_i, f_j), \sum_{(f_i, f_j) \in \hat{A}} \omega_{E^\diamond}^U(f_i, f_j) \right], \left[ \sum_{(f_i, f_j) \in \hat{A}} \nu_{E^\diamond}^L(f_i, f_j), \sum_{(f_i, f_j) \in \hat{A}} \nu_{E^\diamond}^U(f_i, f_j) \right] \right\rangle
$$

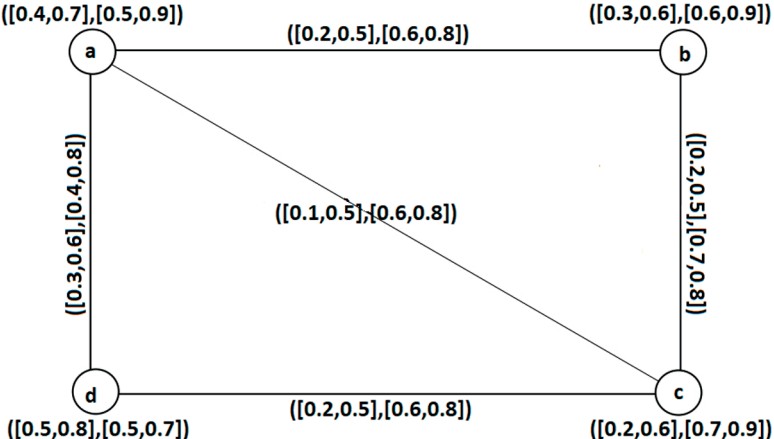

**Fig 2. An IVq-ROFGs $G^\circ$ for a SNC.**

**Table 2. Determinations of the FW of SNC number.**

| $Z$ | $\check{W}_{nc}^{L\omega}(Z)$ | $\check{W}_{nc}^{U\omega}(Z)$ | $\check{W}_{nc}^{L\nu}(Z)$ | $\check{W}_{nc}^{U\nu}(Z)$ |
|---|---|---|---|---|
| $\{a,c\}$ | 0.1+0.1 | 0.5+0.5 | 0.6+0.7 | 0.8+0.8 |
| $\{a,b,c\}$ | 0.1+0.2+0.1 | 0.5+0.5+0.5 | 0.6+0.7+0.7 | 0.8+0.8+0.8 |
| $\{c,d,a\}$ | 0.1+0.2+0.1 | 0.5+0.5+0.5 | 0.7+0.6+0.6 | 0.8+0.8+0.8 |
| $\{a,c,d\}$ | 0.1+0.1+0.2 | 0.5+0.5+0.5 | 0.6+0.7+0.6 | 0.8+0.8+0.8 |
| $\{b,a,c\}$ | 0.2+0.1+0.1 | 0.5+0.5+0.5 | 0.7+0.6+0.7 | 0.8+0.8+0.8 |
| $\{a,b,c,d\}$ | 0.1+0.2+0.1+0.2 | 0.5+0.5+0.5+0.5 | 0.6+0.7+0.7+0.6 | 0.8+0.8+0.8+0.8 |

**Table 3. FWs of SNC number.**

| $Z$ | $\check{W}_{nc}(Z)$ |
|---|---|
| $\{a,c\}$ | $\langle[0.2,1.0],[1.3,1.6]\rangle$ |
| $\{a,b,c\}$ | $\langle[0.4,1.5],[2.0,2.4]\rangle$ |
| $\{c,d,a\}$ | $\langle[0.4,1.5],[1.9,2.4]\rangle$ |
| $\{a,c,d\}$ | $\langle[0.4,1.5],[1.9,2.4]\rangle$ |
| $\{b,a,c\}$ | $\langle[0.4,1.5],[2.0,2.4]\rangle$ |
| $\{a,b,c,d\}$ | $\langle[0.6,2.0],[2.6,3.2]\rangle$ |

where $\omega_{E\diamond}^{L}(f_i,f_j)$ and $\omega_{E\diamond}^{U}(f_i,f_j)$ denote the min of the lower degree of membership and upper degree of membership of all SAs and $\nu_{E\diamond}^{L}(f_i,f_j)$ and $\nu_{E\diamond}^{U}(f_i,f_j)$ represent the max of the lower degree of non-membership and upper degree of non-membership of SAs.

A set $\lambda_1(H) = \lambda_1 = \langle[\lambda_{11}^{L\omega},\lambda_{11}^{U\omega}],[\lambda_{21}^{L\nu},\lambda_{21}^{U\nu}]\rangle$ is an SAC number of IVq-ROFGs, if

$$\lambda_{11}^{L\omega} = min\{\check{W}_{ac}^{L\omega}(\hat{A})|\hat{A} \text{ is SAC of } G^\circ\}$$

$$\lambda_{11}^{U\omega} = min\{\check{W}_{ac}^{U\omega}(\hat{A})|\hat{A} \text{ is SAC of } G^\circ\}$$

$$\lambda_{21}^{L\nu} = max\{\check{W}_{ac}^{L\nu}(\hat{A})|\hat{A} \text{ is SAC of } G^\circ\}$$

$$\lambda_{21}^{U\nu} = max\{\check{W}_{ac}^{U\nu}(\hat{A})|\hat{A} \text{ is SAC of } G^\circ\}$$

SAC with min degree of membership and max degree of non-membership is said to be a min SAC in IVq-ROFGs $G^\circ$.

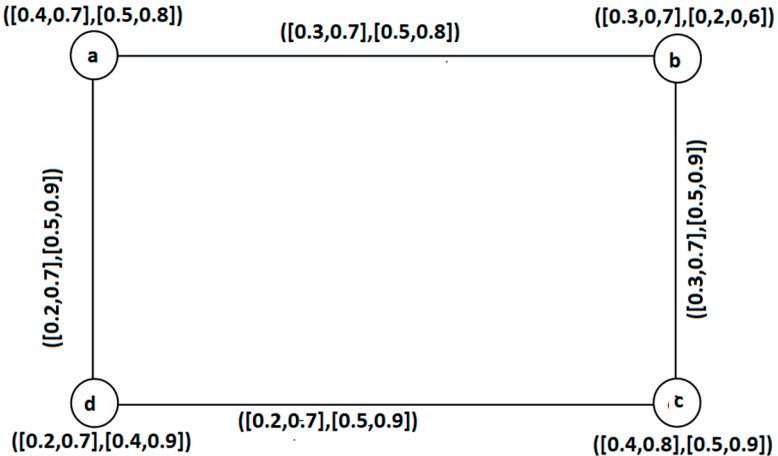

**Fig 3. A strong IVq-ROFGs $G^\circ$ for a SIS.**

**Table 4. FWs of SI number.**

| $Z$ | $\check{W}_{is}^{L\omega}(Z)$ | $\check{W}_{is}^{U\omega}(Z)$ | $\check{W}_{is}^{L\nu}(Z)$ | $\check{W}_{is}^{U\nu}(Z)$ | $\check{W}_{is}(Z)$ |
|---|---|---|---|---|---|
| $\{a,c\}$ | 0.2+0.2 | 0.7+0.7 | 0.5+0.5 | 0.9+0.9 | $\langle[0.4,1.4],[1.0,1.8]\rangle$ |
| $\{b,d\}$ | 0.3+0.2 | 0.7+0.7 | 0.5+0.5 | 0.9+0.9 | $\langle[0.5,1.4],[1.0,1.8]\rangle$ |

**Proposition 37.** *If $\dot{G}$ be an IVq–ROFC such that $\vec{G}$ be a cycle with $\hat{n}$ number of vertices in $\dot{G}$, then*

$$\lambda_{11}^{L\omega} = min\{\check{W}_{ac}^{L\omega}(\hat{A})|\hat{A} \text{ is SAC of } \dot{G} \text{ with } |\hat{A}| \geq \lceil\tfrac{\hat{n}}{2}\rceil\}$$

$$\lambda_{11}^{U\omega} = min\{\check{W}_{ac}^{U\omega}(\hat{A})|\hat{A} \text{ is SAC of } \dot{G} \text{ with } |\hat{A}| \geq \lceil\tfrac{\hat{n}}{2}\rceil\}$$

$$\lambda_{21}^{L\nu} = max\{\check{W}_{ac}^{L\nu}(\hat{A})|\hat{A} \text{ is SAC of } \dot{G} \text{ with } |\hat{A}| \geq \lceil\tfrac{\hat{n}}{2}\rceil\}$$

$$\lambda_{21}^{U\nu} = max\{\check{W}_{ac}^{U\nu}(\hat{A})|\hat{A} \text{ is SAC of } \dot{G} \text{ with } |\hat{A}| \geq \lceil\tfrac{\hat{n}}{2}\rceil\}$$

*Proof*: In an IVq-ROFC $\dot{G}$, all of the arcs are SAs. Due to this, the no. of edges in SAC of both $\dot{G}$ and $\vec{G}$ are similar. also, the SAC number of $\vec{G}$ is $\lceil\tfrac{\hat{n}}{2}\rceil\}$. Hence, $\lceil\tfrac{\hat{n}}{2}\rceil\}$ is the min number of edges in SAC of $\dot{G}$. □

**Theorem 38.** Let $G^\circ$ is a complete IVq-ROFGs. Then

$$\lambda_{11}^{L\omega} = min\{\check{W}_{ac}^{L\omega}(\hat{A})|\hat{A} \text{ is SAC of } G^\circ \text{ with } |\hat{A}| \geq \lceil\tfrac{\hat{n}}{2}\rceil\}$$

$$\lambda_{11}^{U\omega} = min\{\check{W}_{ac}^{U\omega}(\hat{A})|\hat{A} \text{ is SAC of } G^\circ \text{ with } |\hat{A}| \geq \lceil\tfrac{\hat{n}}{2}\rceil\}$$

$$\lambda_{21}^{L\nu} = max\{\check{W}_{ac}^{L\nu}(\hat{A})|\hat{A} \text{ is SAC of } G^\circ \text{ with } |\hat{A}| \geq \lceil\tfrac{\hat{n}}{2}\rceil\}$$

$$\lambda_{21}^{U\nu} = max\{\check{W}_{ac}^{U\nu}(\hat{A})|\hat{A} \text{ isSAC of } G^\circ \text{ with } |\hat{A}| \geq \lceil\tfrac{\hat{n}}{2}\rceil\}$$

*in $G^\circ$, $\hat{n}$ represents the number of nodes.*

*Proof*: Let $G^\circ$ be a complete IVq-ROFGs. Then every arc of complete IVq-ROFGs is SAs and all vertices are connected to each other. Since all of the arcs are SAs in an IVq-ROFGs $G^\circ$

and crisp graph $\dot{G}$. Then $\dot{G}$ and $G^\circ$ have the equal no. of edges. Now, $\lceil \frac{\hat{n}}{2} \rceil$ is the SAC number of the crisp graph $\dot{G}$. Thus $\lceil \frac{\hat{n}}{2} \rceil$ is the minimum numbers of arc in SAC of $G^\circ$. □

**Theorem 39.** *if a complete bipartite IVq-ROFGs $\check{G}$ is partitioned into two subsets $V_1^\diamond$ and $V_2^\diamond$. Then*

$$\lambda_{11}^{L\omega}(\check{G}) = min\{\check{W}_{ac}^{L\omega}(\hat{A})|\hat{A} \text{ is SAC in } \check{G} \text{ with } |\hat{A}| \geq max\{|V_1^\diamond|, |V_2^\diamond|\}$$

$$\lambda_{11}^{U\omega}(\check{G}) = min\{\check{W}_{ac}^{U\omega}(\hat{A})|\hat{A} \text{ is SAC in } \check{G} \text{ with } |\hat{A}| \geq max\{|V_1^\diamond|, |V_2^\diamond|\}$$

$$\lambda_{21}^{L\nu}(\check{G}) = max\{\check{W}_{ac}^{L\nu}(\hat{A})|\hat{A} \text{ is SAC in } \check{G} \text{ with } |\hat{A}| \geq max\{|V_1^\diamond|, |V_2^\diamond|\}$$

$$\lambda_{21}^{U\nu}(\check{G}) = max\{\check{W}_{ac}^{U\nu}(\hat{A})|\hat{A} \text{ is SAC in } \check{G} \text{ with } |\hat{A}| \geq max\{|V_1^\diamond|, |V_2^\diamond|\}$$

*Proof*: Consider that $\check{G}$ is a complete bipartite IVq-ROFGs. Then every arc of complete bipartite IVq-ROFGs is SAs and all of the nodes in $V_1^\diamond$ are linked to all other nodes in $V_2^\diamond$. Since all of the arcs are SAs in IVq-ROFGs $G^\circ$ and crisp graph $\dot{G}$, thus, both the graphs have same numbers of SAC. The SAC number of $G^\circ$ is $max\{|V_1^\diamond|, |V_2^\diamond|\}$. Thus, the min number of arcs is $max\{|V_1^\diamond|, |V_2^\diamond|\}$ is the SAC of $\check{G}$. □

**Definition 40.** Let $G^\circ = (V^\diamond, E^\diamond)$ be an IVq-ROFGs. The collection $\mathring{A}$ of SACs is called a SIS, if any two arcs do not share a vertex. The set $\mathring{A}$ is also called a SM in $G^\circ$.

**Definition 41.** Let $\mathring{A}$ be an SM in IVq-ROFGs. If $(f_i, f_j) \in \mathring{A}$, then $f_i$ is said to be matched strongly to $f_j$. The IVq-ROFW $\check{W}$ of SM $\mathring{A}$ can be described as

$$\check{W}_{sm}(\mathring{A}) = \left\langle \left[ \check{W}_{sm}^{L\omega}(\mathring{A}), \check{W}_{sm}^{U\omega}(\mathring{A}) \right], \left[ \check{W}_{sm}^{L\nu}(\mathring{A}), \check{W}_{sm}^{U\nu}(\mathring{A}) \right] \right\rangle$$

$$\check{W}_{sm}(\mathring{A}) = \left\langle \left[ \sum_{(f_i, f_j) \in \mathring{A}} \omega_{E^\diamond}^L(f_i, f_j), \sum_{(f_i, f_j) \in \mathring{A}} \omega_{E^\diamond}^U(f_i, f_j) \right], \left[ \sum_{(f_i, f_j) \in \mathring{A}} \nu_{E^\diamond}^L(f_i, f_j), \sum_{(f_i, f_j) \in \mathring{A}} \nu_{E^\diamond}^U(f_i, f_j) \right] \right\rangle$$

A collection $\varsigma_1(G^\circ) = \varsigma_1 = \langle [\varsigma_{11}^{L\omega}, \varsigma_{11}^{U\omega}], [\varsigma_{21}^{L\nu}, \varsigma_{21}^{U\nu}] \rangle$ is an SM in IVq-ROFGs, if

$$\varsigma_{11}^{L\omega} = max\{\check{W}_{is}^{L\omega}(\mathring{A})|\mathring{A} \text{ is the SM of } G^\circ\}$$

$$\varsigma_{11}^{U\omega} = max\{\check{W}_{is}^{U\omega}(\mathring{A})|\mathring{A} \text{ is the SM of } G^\circ\}$$

$$\varsigma_{21}^{L\nu} = min\{\check{W}_{is}^{L\nu}(\mathring{A})|\mathring{A} \text{ is the SM of } G^\circ\}$$

$$\varsigma_{21}^{U\nu} = min\{\check{W}_{is}^{U\nu}(\mathring{A})|\mathring{A} \text{ is the SM of } G^\circ\}$$

SM with max degree of membership and min degree of non-membership is called max SM in IVq-ROFGs $G^\circ$.

**Proposition 42.** *if $\dot{G}$ be an IVq-ROFC such that $\vec{G}$ be a cycle with $\hat{n}$ number of vertices in $\dot{G}$, then*

$$\varsigma_{11}^{L\omega} = max\{\check{W}_{is}^{L\omega}(\mathring{A})|\mathring{A} \text{ is the SM of } \dot{G} \text{ with } |\mathring{A}| \leq \lceil \frac{\hat{n}}{2} \rceil\}$$

$$\varsigma_{11}^{U\omega} = max\{\check{W}_{is}^{U\omega}(\mathring{A})|\mathring{A}\mathring{A} \text{ is the SM of } \dot{G} \text{ with } |\mathring{A}| \leq \lceil \frac{\hat{n}}{2} \rceil\}$$

$$\varsigma_{21}^{L\nu} = min\{\check{W}_{is}^{L\nu}(\mathring{A})|\mathring{A}\mathring{A} \text{ is the SM of } \dot{G} \text{ with } |\mathring{A}| \leq \lceil \frac{\hat{n}}{2} \rceil\}$$

$$\varsigma_{21}^{U\nu} = min\{\check{W}_{is}^{U\nu}(\mathring{A})|\mathring{A}\mathring{A} \text{ is the SM of } \dot{G} \text{ with } |\mathring{A}| \leq \lceil \frac{\hat{n}}{2} \rceil\}$$

*Proof*: In an IVq-ROFC $\dot{G}$, all of the arcs are SAs. Due to this, the no. of edgs in SM of $\dot{G}$ and $\vec{G}$ are similer. Now, $\lceil \frac{\hat{n}}{2} \rceil\}$ is the SM number of $\vec{G}$. Thus, $\lceil \frac{\hat{n}}{2} \rceil\}$ is also the max number of edges in SM of $\dot{G}$. □

**Theorem 43.** *Let $G°$ be a complete IVq-ROFGs. Then*

$$\varsigma_{11}^{L\omega} = max\{\breve{W}_{sm}^{L\omega}(Å)|Å \text{ is SM with } |Å| \leq \lceil \tfrac{\hat{n}}{2} \rceil\}$$

$$\varsigma_{11}^{U\omega} = max\{\breve{W}_{sm}^{U\omega}(Å)|Å \text{ is SM with } |Å| \leq \lceil \tfrac{\hat{n}}{2} \rceil\}$$

$$\varsigma_{21}^{L\nu} = min\{\breve{W}_{sm}^{L\nu}(Å)|Å \text{ is SM with } |Å| \leq \lceil \tfrac{\hat{n}}{2} \rceil\}$$

$$\varsigma_{21}^{U\nu} = min\{\breve{W}_{sm}^{U\nu}(Å)|Å \text{ is SM with } |Å| \leq \lceil \tfrac{\hat{n}}{2} \rceil\}$$

*where in $G°$, $\hat{n}$ represents the number of nodes.*

*Proof*: Let $G°$ be a complete IVq-ROFGs. Then, every arc of complete IVq-ROFGs is the SAs and all the nodes are connected to each other. Since all of the arcs are SAs in a crisp graph $\dot{G}$ and IVq-ROFGs $G°$, Thus, in SM the graphs $\dot{G}$ and $G°$ have equal numbers of arcs. In crisp graph $\dot{G}$ the number of SM is represented by $\lceil \tfrac{\hat{n}}{2} \rceil$. Thus, the max number of arcs in SM of $G°$ is $\lceil \tfrac{\hat{n}}{2} \rceil$. □

**Theorem 44.** *If a complete bipartite IVq-ROFGs $\breve{G}$ divided into subsets $V_1^\diamond$ and $V_2^\diamond$, then*

$$\varsigma_{11}^{L\omega}(\breve{G}) = max\{\breve{W}_{sm}^{L\omega}(Å)|Å \text{ is SM in } \breve{G} \text{ with } |Å| \leq min\{|V_1^\diamond|, |V_2^\diamond|\}\}$$

$$\varsigma_{11}^{U\omega}(\breve{G}) = max\{\breve{W}_{sm}^{U\omega}(Å)|Å \text{ is SM in } \breve{G} \text{ with } |Å| \leq min\{|V_1^\diamond|, |V_2^\diamond|\}\}$$

$$\varsigma_{21}^{L\nu}(\breve{G}) = min\{\breve{W}_{sm}^{L\nu}(Å)|Å \text{ is SM in } \breve{G} \text{ with } |Å| \leq min\{|V_1^\diamond|, |V_2^\diamond|\}\}$$

$$\varsigma_{21}^{U\nu}(\breve{G}) = min\{\breve{W}_{sm}^{U\nu}(Å)|Å \text{ is SM in } \breve{G} \text{ with } |Å| \leq min\{|V_1^\diamond|, |V_2^\diamond|\}\}$$

*Proof*: Let $\breve{G}$ be a complete bipartite IVq-ROFGs. Then every arc of complete bipartite IVq-ROFGs is the SAs and all of the vertices in $V_1^\diamond$ are linked to every other vertices in $V_2^\diamond$. Since all of the arcs are SAs in IVq-ROFGs $\breve{G}$ and crisp graph $\dot{G}$, hence both of the graphs has equal amount of arcs in SM. Since $max\{|V_1^\diamond|, |V_2^\diamond|\}$ is the SM number of $\breve{G}$. Thus, the max number of arcs in SM of $\breve{G}$ is $min\{|V_1^\diamond|, |V_2^\diamond|\}$. □

**Example 45.** Refer to IVq-ROFGs given in Fig 4. Here, all of the arcs are SAs and all the SACs of IVq-ROFGs $G°$ are: $\hat{A}_1 = \{ab, cd\}$, $\hat{A}_2 = \{ad, bc\}$, $\hat{A}_3 = \{ba, ac, cd\}$, $\hat{A}_4 = \{ac, cb, cd\}$, $\hat{A}_5 = \{ac, ab, ad\}$, $\hat{A}_6 = \{ab, ad, dc\}$, $\hat{A}_7 = \{ad, ab, bc\}$, $\hat{A}_8 = \{ad, dc, cb\}$. The FW of SAC number $\breve{W}_{ac}(\hat{A})$ of IVq-ROFGs $G°$ is calculated in Table 5. Hence, $\lambda_1(G°) = \langle[0.5, 1.2], [2.1, 2.7]\rangle$. Also, the only SM and SAC in $G°$ are these two sets $\hat{A}_1$ and $\hat{A}_2$. Thus $\breve{W}_{sm}(\hat{A}) = \langle[0.5, 1.2], [1.3, 1.8]\rangle$, $\breve{W}_{sm}(\hat{A}) = \langle[0.6, 1.3], [1.2, 1.8]\rangle$. Thus, $\varsigma_1 = \langle[0.6, 1.3], [1.2, 1.8]\rangle$.

**Example 46.** Suppose an IVq-ROFGs $G°$ as shown in Fig 2. Clearly, $ad, cd$ and $bc$ are its SAs and all of the SACs of $G°$ are: $\hat{A}_1 = \{ad, bc\}$, $\hat{A}_2 = \{ad, cd, bc\}$ The IVq-ROFW of SAC number of IVq-ROFGs $G°$ is calculated in Table 6. Thus, $\lambda_1(G°) = \langle[0.5, 1.4], [1.5, 2.7]\rangle$. The set $\hat{A}_1 = \{ad, bc\}$ is the only SIAC. Hence, $\varsigma_1(G°) = \langle[0.5, 1.4], [1.0, 1.8]\rangle$.

**Definition 47.** Let $Å$ be an SM in IVq-ROFGs $G°$. Then, $Å$ is called PSM, if all of the vertices of $Å$ in $G°$ are strongly matched to some vertices of $G°$.

**Example 48.** Consider an IVq-ROFGs depicted in Fig 5. All of its arcs are SAs and the collections $N_1$ and $N_2$ are called PSM. All SM of $G°$ are: $N_1 = \{ad, bc\}$, $N_2 = \{ac, bd\}$, $N_3 =$

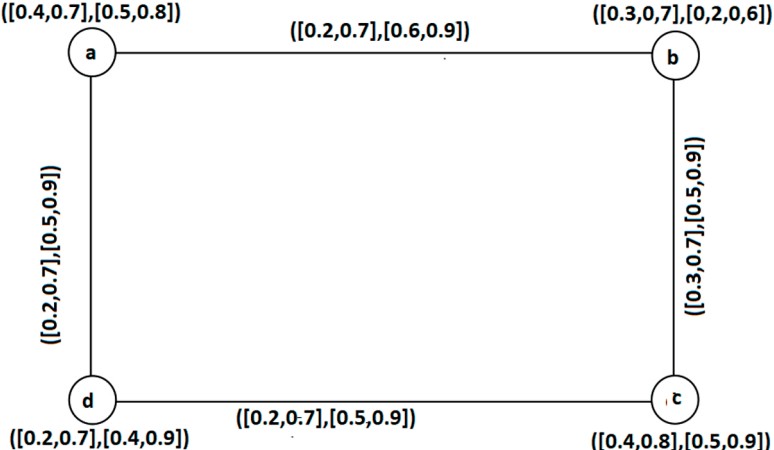

**Fig 4. An IVq-ROFGs $G°$ for SIAC.**

**Table 5. Determinations of the FWs of SAC number.**

| $\hat{A}$ | $\breve{W}_{ac}^{L\omega}(\hat{A})$ | $\breve{W}_{ac}^{U\omega}(\hat{A})$ | $\breve{W}_{ac}^{L\nu}(\hat{A})$ | $\breve{W}_{ac}^{U\nu}(\hat{A})$ | $\breve{W}_{ac}(\hat{A})$ |
|---|---|---|---|---|---|
| $\hat{A}_1$ | 0.3+0.2 | 0.6+0.6 | 0.6+0.7 | 0.9+0.9 | $\langle[0.5,1.2],[1.3,1.8]\rangle$ |
| $\hat{A}_2$ | 0.4+0.2 | 0.7+0.6 | 0.5+0.7 | 0.9+0.9 | $\langle[0.6,1.3],[1.2,1.8]\rangle$ |
| $\hat{A}_3$ | 0.3+0.2+0.2 | 0.6+0.6+0.6 | 0.6+0.7+0.7 | 0.9+0.9+0.9 | $\langle[0.7,1.8],[2.0,2.7]\rangle$ |
| $\hat{A}_4$ | 0.2+0.2+0.2 | 0.6+0.6+0.6 | 0.7+0.7+0.7 | 0.9+0.9+0.9 | $\langle[0.6,1.8],[2.1,2.7]\rangle$ |
| $\hat{A}_5$ | 0.2+0.3+0.4 | 0.6+0.6+0.7 | 0.7+0.6+0.5 | 0.9+0.9+0.9 | $\langle[0.9,1.9],[1.8,2.7]\rangle$ |
| $\hat{A}_6$ | 0.3+0.4+0.2 | 0.6+0.7+0.6 | 0.6+0.5+0.7 | 0.9+0.9+0.9 | $\langle[0.9,1.9],[1.8,2.7]\rangle$ |
| $\hat{A}_7$ | 0.4+0.3+0.2 | 0.7+0.6+0.6 | 0.5+0.6+0.7 | 0.9+0.9+0.9 | $\langle[0.9,1.9],[1.8,2.7]\rangle$ |
| $\hat{A}_8$ | 0.4+0.2+0.2 | 0.7+0.6+0.6 | 0.5+0.7+0.7 | 0.9+0.9+0.9 | $\langle[0.8,1.9],[1.9,2.7]\rangle$ |

**Table 6. Calculations of the FWs of SAC set.**

| $\hat{A}$ | $\breve{W}_{ac}^{L\omega}(\hat{A})$ | $\breve{W}_{ac}^{U\omega}(\hat{A})$ | $\breve{W}_{ac}^{L\nu}(\hat{A})$ | $\breve{W}_{ac}^{U\nu}(\hat{A})$ | $\breve{W}_{ac}(\hat{A})$ |
|---|---|---|---|---|---|
| $\hat{A}_1$ | 0.2+0.3 | 0.7+0.7 | 0.5+0.5 | 0.9+0.9 | $\langle[0.5,1.4],[1.0,1.8]\rangle$ |
| $\hat{A}_2$ | 0.2+0.2+0.3 | 0.7+0.7+0.7 | 0.5+0.5+0.5 | 0.9+0.9+0.9 | $\langle[0.7,2.1],[1.5,2.7]\rangle$ |

$\{ad, ac, bc\}$, $N_4 = \{ad, db, bc\}$. The IVq-ROFW of SM is calculated in Table 7. Hence, $\lambda_1(N) = \langle[0.5,1.2],[1.9,2.6]\rangle$, $\varsigma_1(N) = \langle[0.9,1.8],[1.2,1.7]\rangle$.

## 5 Analysis of Fuzzy Air Conditioning System (FACS)

An air conditioner (AC) is an electrical device that ustilizes air conditioning system and helps to keep the temperature of the surroundings according to the need. The AC system starts when the thermostat detects that the indoor temperature is greater than the set (targeted) temperature. The compressor begins to circulate refrigerant that absorbs the heat from indoor air in the evaporator coil. The heated refrigerant gas is pumped towards the condenser coil, where it releases the heat in the outside air and condenses back into a liquid. Afterwards, the indoor air is circulated by the blower fan over the evaporator coil which cools and dehumidifies this air. Finally, the ductwork distributes the cool air into the room or building. The AC

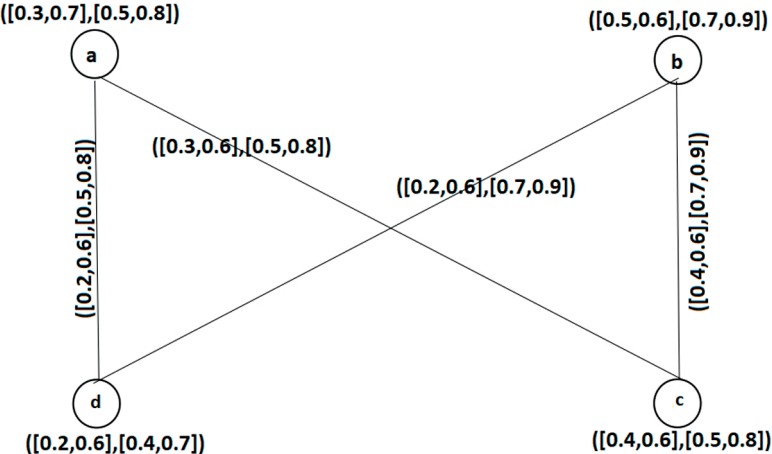

**Fig 5. An IVq-ROFGs $G^\circ$ for SM.**

**Table 7. Calculations of the FWs of SM.**

| $N$ | $\check{W}_{sm}^{L\omega}(N)$ | $\check{W}_{sm}^{U\omega}(N)$ | $\check{W}_{sm}^{L\nu}(N)$ | $\check{W}_{sm}^{U\nu}(N)$ | $\check{W}_{sm}(N)$ |
|---|---|---|---|---|---|
| $N_1$ | 0.2+0.4 | 0.6+0.6 | 0.5+0.7 | 0.8+0.9 | $\langle[0.6, 1.2], [1.2, 1.7]\rangle$ |
| $N_2$ | 0.3+0.2 | 0.6+0.6 | 0.5+0.7 | 0.8+0.9 | $\langle[0.5, 1.2], [1.2, 1.7]\rangle$ |
| $N_3$ | 0.2+0.3+0.4 | 0.6+0.6+0.6 | 0.5+0.5+0.7 | 0.8+0.8+0.9 | $\langle[0.9, 1.8], [1.7, 2.5]\rangle$ |
| $N_4$ | 0.2+0.2+0.4 | 0.6+0.6+0.6 | 0.5+0.7+0.7 | 0.8+0.9+0.9 | $\langle[0.8, 1.8], [1.9, 2.6]\rangle$ |

system turned off automatically as the set temperature on the thermostat was achieved. It cycles on again when the temperature rises above the set point.

Usually, air conditioning system depends on binary decisions (on/off) based on temperature. However, FACS works on fuzzy logic that allows more flexible and adaptable control. An air conditioning system is designed to maintain a particular temperature range while the FACS has the capability of obtaining precise and adaptable cooling responses based on varying conditions. FACS makes AC to maintain the indoor temperature by comparing the room temperature and the targeted temperature. However, sometimes we encounter variations in temperature at different points within a large hall. In such circumstances, we consider a hall divided in to sections (portions) to handle this complex situation. To detect temperature in different sections of a hall, we use the temperature sensors $\{a, b, c, d, e\}$. Sometimes, the temperature readings from two different sensors are either identical or interdependent, and the temperature in the area between the two sensors is influenced by both reading. In such cases, we model such types of complex situations using the concepts of covering and matching in IVq-ROFGs. Specifically, we address these circumstances using the concepts of SISs, SNC etc in IVq-ROFGs. We investigate the effectiveness or ineffectiveness of the temperature levels in comparison to the targeted temperature in different sections. Our aim is to identify the sections of the hall where the temperature levels are ineffective or fall outside the specified range. We evaluate these by utilizing the concept of SISs of covering in IVq-ROFGs. Let us consider that the temperature sensors $\{a, b, c, d, e\}$ placed in different sections of the hall are represented by the vertices that detect temperature levels at different points while the edges denote the realtionship between two sensors as depicted in Fig 6. In IVq-ROFGs, membership

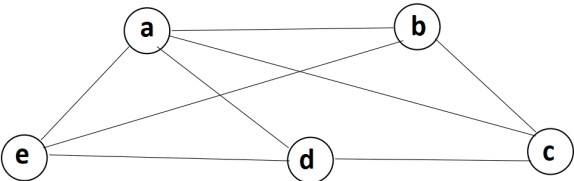

**Fig 6. An IVq-ROFGs of Temperature sensors.**

degree of vertices and edges depict the effectiveness of the temperatures levels, while the non-membership degrees of vertices and edges depict their ineffectiveness. By effectiveness of the temperature level, we mean that the temperature is either close to the desired value or there is only a slight differences between hall's temperature and the targeted temperature. Similarly, ineffectiveness of the temperature level indicates that the temperature does not meet our requirements or is completely unrelated. The degrees of membership and non-membership corresponding to each temperature sensor (vertex) are shown in Table 8 while the degrees of membership and non-membership of each edge are presented in Table 9. By utilizing the concept of $M$SIS of covering in IVq-ROFGs, we detect the effectiveness or ineffectiveness of the temperature levels in those sections of the hall where the current temperature doesn't meet our requirements. The $M$SIS from the collection of SISs represents the sections where the temperature levels are nearly ineffective.

**Computational framework.** First, we find the SISs and then we find the $M$SIS from the collection of SISs. By utilizing the proposed algorithm 1, we identify the SISs (see Table 10) of an IVq-ROFGs shown in Fig 6.

**Algorithm 1:**

1. Suppose $s$ be any vertex in $X$. Remove all the vertices adjacent to $s$.
2. Now suppose any other vertex in the remaining graph belongs to $X$. Hence different IS, including $s$ can be obtained depending on the vertex selected from the remaining of the graph and in $X$.
3. For selection of all possible vertices repeat step 2.

Further to the above, $M$SIS is calculated on the basis of fuzzy weights (FWs), as the unique SISs are $Z_1 = \{b, d\}$ and $Z_2 = \{c, e\}$. The IVq-ROFWs corresponding to the sets given above are $\breve{W}(Z_1) = \langle [1.1, 1.6], [0.8, 1.4] \rangle$ and $\breve{W}(Z_2) = \langle [0.7, 1.3], [1.0, 1.6] \rangle$. Clearly, $\breve{W}(Z_1)$ has max degree of membership and min degree of non-membership as compared to $\breve{W}(Z_2)$. This implies that $Z_1$ is selected as $M$SISs from the Table 10 on the base of its FWs. The $M$SIS $Z_1 = \{b, d\}$ represents those temperature sensors whose detected temperature is totally different or independent. Furthermore, in between these two sections of a hall that have the temperature sensors $b$ and $d$, we can see that there is a varying condition of temperature. In general, we can say that the temperature at sensor $b$ is cold and the temperature at sensor $d$ is hot. Then on the basis of this condition, we can adjust the FACS according to the need of such sections. In this regard, the following five fuzzy logic levels can be considered i.e., the temperature level $[45°C – 55°C]$ is very hot (VH), $[35°C – 45°C]$ is hot (H), $[25°C – 35°C]$ is warm (W), $[15°C – 25°C]$ is cold (C) and $[0°C – 15°C]$ is very cold (VC). Basically, FACS maintains the temperature based on three commands i.e., heat (HE), cool (CO) and no change (NC). Fig 7 manipulates the FACS according to the fuzzy logic levels (i.e., VC, C, W, H, VH) and commands (i.e., HE, CO and NC). In Fig 8, the flow chart elaborates the

**Table 8. The values for vertices of IVq-ROFGs of the Temperature sensors.**

| Vertices | Temp–level | IVq-ROFGs corresponding to each section |
|---|---|---|
| $a$ | very cold | $\langle [0.8, 1.0], [0.3, 0.6] \rangle$ |
| $b$ | cold | $\langle [0.7, 0.9], [0.2, 0.5] \rangle$ |
| $c$ | warm | $\langle [0.6, 0.8], [0.7, 0.9] \rangle$ |
| $d$ | hot | $\langle [0.4, 0.7], [0.5, 0.8] \rangle$ |
| $e$ | very hot | $\langle [0.1, 0.5], [0.3, 0.7] \rangle$ |

**Table 9. The values for edges of IVq-ROFGs of the Temperature sensors.**

| Edges | IVq-ROFGs | Edges | IVq-ROFGs |
|---|---|---|---|
| $ab$ | $\langle [0.7, 0.9], [0.3, 0.6] \rangle$ | $ae$ | $\langle [0.1, 0.5], [0.3, 0.7] \rangle$ |
| $bc$ | $\langle [0.6, 0.8], [0.7, 0.9] \rangle$ | $ac$ | $\langle [0.6, 0.8], [0.7, 0.9] \rangle$ |
| $cd$ | $\langle [0.4, 0.7], [0.7, 0.9] \rangle$ | $ad$ | $\langle [0.4, 0.7], [0.5, 0.8] \rangle$ |
| $de$ | $\langle [0.1, 0.5], [0.5, 0.8] \rangle$ | $be$ | $\langle [0.1, 0.5], [0.3, 0.7] \rangle$ |

**Table 10. The collection of the SISs.**

| Step1 | Step2 | SISs |
|---|---|---|
| $b$ | $d$ | $\{b, d\}$ |
| $c$ | $e$ | $\{c, e\}$ |
| $d$ | $b$ | $\{d, b\}$ |
| $e$ | $c$ | $\{e, c\}$ |

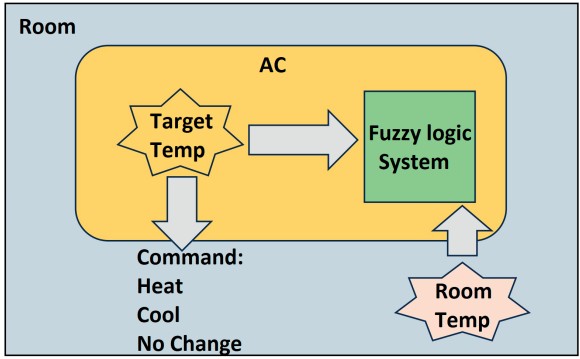

**Fig 7. FACS.**

working of FACS. Furthermore, the working of FACS for all fuzzy logic levels is presented in Table 11.

Our computational frame work comprises of algorithm 1 and pseudocode 1 establishes the facts about our proposed methodology. For further elaboration of our proposed model, we present two numerical examples in Sect 6 and verify this model is more useful to express the complex scenario compared to other models presented through IFGs and PyFGs.

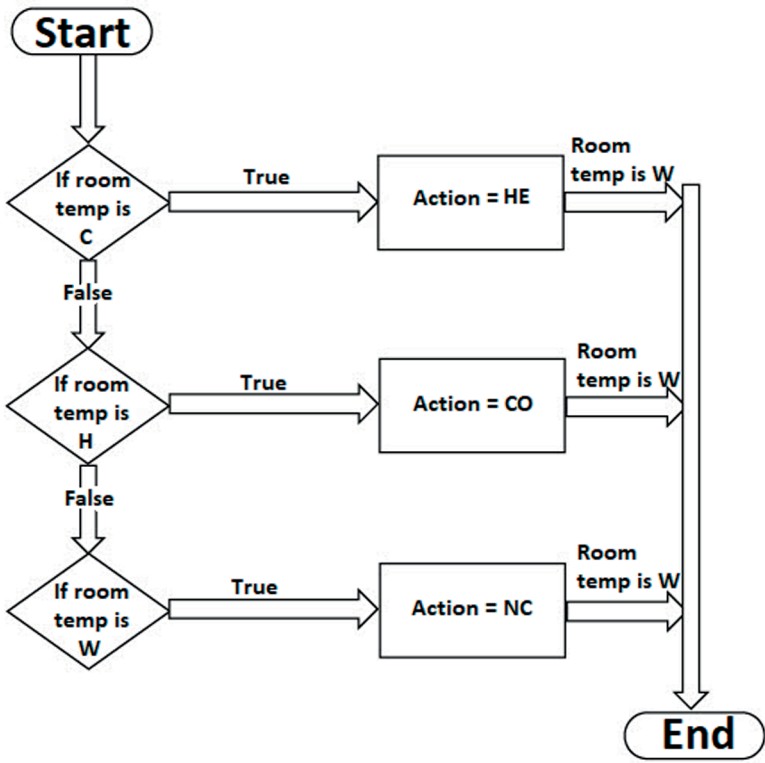

**Fig 8. Flow chart of the working of FACS.**

**Table 11. A working of FACS.**

| Room temp/Target temp | Very cold | Cold | Warm | Hot | Very hot |
|---|---|---|---|---|---|
| Very cold | NC | HE | HE | HE | HE |
| Cold | CO | NC | HE | HE | HE |
| Warm | CO | CO | NC | HE | HE |
| Hot | CO | CO | CO | NC | HE |
| Very hot | CO | CO | CO | CO | NC |

**Pseudocode 1:**

1. **Initialize Variables**
   ◇ Create a list of sensors: $sensors = [a, b, c, d, e]$
   ◇ Define the target temperature range: $target - temp = (20, 25)$
   ◇ Input temperature readings: $temp - readings = [30, 22, 15, 28, 40]$
2. **For Loop to Identify Sensors Outside the Target Range**
   **For** $(sensor = 0, sensor < 5, sensor + +)$:
   If the temperature of the current sensor is outside the target range:
   Add the sensor and its temperature to outside-sensors.
3. **Define Fuzzy Membership Functions for Temperature**
   ◇ Define membership functions for temperature:
   · low for temperatures $[0, 20]$
   · ok for temperatures $[20, 28]$
   · high for temperatures $[25, 50]$

4. **Define Fuzzy Membership Functions for Heating and Cooling**
5. **Define Fuzzy Control Rules**
6. **Create Fuzzy Control Systems**
   ◇ Initialize cooling-ctrl for cooling logic.
   ◇ Initialize heating-ctrl for heating logic.
7. **Process Each Sensor Using a For Loop**
   **For** $(sensor = 0, sensor < 5, sensor + +)$:
   Retrieve the current temperature: $temp - reading = temp - readings[sensor]$.
   **If** $temp - reading < target - temp[0]$ $(temperature\,is\,below\,range)$:
   · Apply the heating-ctrl control system.
   · Set the input to the sensor's temperature.
   · Compute the heating output.
   · Display: Heating activated for sensor $\{sensors[sensor]\}$ with output $\{heating - output\}$.
   **Else if** $temp - reading > target - temp[1]$ $(temperature\,is\,above\,range)$:
   · Apply the cooling-ctrl control system.
   · Set the input to the sensor's temperature.
   · Compute the cooling output.
   · Display: Cooling activated for sensor $\{sensors[sensor]\}$ with output $\{cooling - output\}$.
   **Else** (temperature is within the target range):
   · Display: No change needed for sensor $\{sensors[sensor]\}$.

## 6 Comparative analysis and superiority of our proposed study

The notion of FGs was introduced by Rosenfeld. FGs use degree of membership for nodes and edges satisfying a condition. By utilizing FGs, many real-world vague problems were resolved and useful results were obtained. In graphs, the notion of covering and matching is very important and has numerous applications. In FGs, several vague problems have been resolved based on covering and matching. Several extensions of FGs were introduced to deals with uncertain problems. The most important extensions of FGs named IVFGs, IFGs and IVIFGs. FGs represents the data by using the degree of membership as a single element from $[0, 1]$ and for IVFGs the degree of membership is in the form of sub-intervals of $[0, 1]$. However, with the passage of time real-world uncertain or vague problems become more complex that cannot be handled through FGs and IVFGs. Therefore, we need a model or a structure which is more flexible then previous one to obtain more accurate and useful outcomes. To tackle such kind of situations, the concept of IFGs was introduced. Let us consider an example that a person is agree or disagree to do work with an organization. The structure of FGs and IVFGs deal only with agree to do work and assign value of a membership from $[0, 1]$ to conclude whether a person is agree or strongly agree. Sometimes we faced a negative opinion (disagree). To deal with both opinions (agree or disagree), we need a structure that has the ability to deal with such cases more precisely. In this regard, the notion of IFGs introduced in literature, IFGs utilizes the degree of membership and degree of non-membership as a single element from $[0, 1]$ for nodes and edges. Here, we can deal with negative opinions (disagree) through the non-membership. With the passage of time, to handle complex problems more precisely, the concept of IVIFGs was introduced. IVIFGs is more accurate structure and has wide range compared to IFGs, FGs and IVFGs to make a decision. An IVIFGs uses degree of membership and degree of non-membership in the form of sub-intervals of $[0, 1]$ for nodes and arcs. Subsequently, the notions of covering and matching in IVIFGs were introduced which become useful to deal complex problems and generating accurate conclusions for them.

To prove the worthiness of our proposed study, we analyze our proposed notion of covering and matching in the domain of IVq-ROFGs with comparison to the notion of covering and matching in the domain of IVIFGs. In this regard, we deal an example by using covering in IVIFGs and IVq-ROFGs. For this, we consider the data set as shown in Table 12. We find the $M$SIS from the SISs of covering in the setting of IVIFGs and IVq-ROFGs. Furthermore, the comparison between the FWs of $M$SIS shows that the notion of covering and matching in IVq-ROFGs is better and more flexible than covering and matching in IVIFGs. Then we have the $M$SISs from the collection of SISs as:

$$\check{W}(Z_1) = \{b, d\} \text{ and } \check{W}(Z_2) = \{c, e\}.$$
$$\check{W}(Z_1) = \langle [0.9, 1.2], [0.5, 0.6] \rangle$$
$$\check{W}(Z_2) = \langle [0.6, 1.0], [0.5, 0.9] \rangle$$

Clearly, $\check{W}(Z_1)$ has the max degree of membership and min degree of non-membership as compared to $\check{W}(Z_2)$. This implies that $Z_1$ is our required $M$SIS of covering in IVIFGs. Furthermore, if we consider the data set as given in Table 8. An IVIFGs are unable to deal with data that given in Table 8. Because of the taken values for membership and non-membership violate the condition $0 \leq \omega_U + \nu_U \leq 1$ of IVIFGs, as shown in Example 49.

**Example 49.**      Assume the value of VC cluster $\omega_U = 0.7$ and $\nu_U = 0.8$ of VC shown in Table 8, then by the condition

$$0 \leq \omega_U + \nu_U \leq 1$$
$$0 \leq 0.7 + 0.8 \nleq 1$$

The above condition is not satisfied, hence we are unable to continue in the domain of IVIFGs.

To handle such situations, one of the most advanced extension of the FSs named q-ROFSs that allows a wide range for the degree of membership and the degree of non-membership was introduced. Furthermore, the concept of q-ROFGs was introduced by using q-ROFR. The q-ROFGs utilize the values of $[0, 1]$ for the degree of membership and degree of non-membership of nodes and edges. Through the q-ROFGs we can obtain better results but these results are less appropriate and adaptable. Consequently, the concept of IVq-ROFGs was introduced in literature having the same structure as IVFGs, IVIFGs and for the degree of membership and degree of non-membership of the nodes and edges utilize the sub-intervals of $[0, 1]$. To make the structure of an IVq-ROFGs more efficient to deal with the big data set, we initiate the notion of covering and matching in IVq-ROFGs. We calculate the FWs of SNCs, SACs and the $M$SIS by utilizing the notion of covering in IVq-ROFGs. Hence, we can continue our work based on covering and matching in IVq-ROFGs by using the data set given

**Table 12. The values of nodes in the domain of IVIFGs for analysis of FACS.**

| Vertices | Temp–level | IVIFGs corresponding to each section |
|---|---|---|
| a | very cold | $\langle [0.6, 0.7], [0.2, 0.3] \rangle$ |
| b | cold | $\langle [0.5, 0.7], [0.1, 0.2] \rangle$ |
| c | warm | $\langle [0.5, 0.6], [0.3, 0.4] \rangle$ |
| d | hot | $\langle [0.4, 0.5], [0.3, 0.4] \rangle$ |
| e | very hot | $\langle [0.1, 0.4], [0.2, 0.5] \rangle$ |

in Table 8. Moreover, we also see in Example 50 that the data set provided in Table 8 can be modeled through IVq-ROFGs. However, we have observed that the IVFGs is unable to deal with the data set given in Table 8.

**Example 50.** Let us consider the values $\omega_U = 0.7$ and $\nu_U = 0.8$ of VC from the Table 8 with $q = 3$, then

$$0 \leq (\omega_U)^q + (\nu_U)^q \leq 1$$
$$0 \leq (0.7)^3 + (0.8)^3 \leq 1$$

Then the above conditions are satisfied and hence the notion of covering in IVq-ROFGs is more appropriate for our model such data.

Since $MSIS$s are $Z_1 = \{b, d\}$ and $Z_2 = \{c, e\}$. Then by utilizing the data set given in Table 8, we get the IVq-ROFW of the sets as $Z_1 = \{b, d\}$ and $Z_2 = \{c, e\}$ as:

$$\breve{W}(Z_1) = \langle [1.1, 1.6], [0.8, 1.4] \rangle$$
$$\breve{W}(Z_2) = \langle [0.7, 1.3], [1.0, 1.6] \rangle.$$

Clearly, $\breve{W}(Z_1)$ has max degree of membership and min degree of non-membership as compared to $\breve{W}(Z_2)$. This implies that $Z_1$ is selected as the $MSIS$ of covering in the domain of IVq-ROFGs. Moreover, Table 13 shows that the set $Z_1$ has the max degree of membership and min degree of non-membership in both domains. However, in Table 13, we also notice that the $MSIS$ $Z_1$ of covering in the domain of IVq-ROFGs is more precise and effective than the $MSIS$ $Z_1$ of covering in the domain of IVIFGs. In Fig 9, we also provide a graphical comparison of calculated $MSIS$ $Z_1$ of covering in the frame of IVFGs and in the domain of IVq-ROFGs on the basis of FWs to validate that the notion of covering and matching is more effective in the realm of IVq-ROFGs to deal complex problems with a wide range of membership and non-membership degree from $[0, 1]$. At the end, we provide the characteristics comparison of IVq-ROFGs with IVFGs, IVIFGs and IVPyFGs in Table 14.

**Discussions.** Finally, we provide the comparison of our application presented in Sect 5 through Table 13 and Fig 9. It is evident that if we utilize $MSIS$ of covering in the setting of IVFGs instead of IVq-ROFGs, then we are unable to compare the temperature of sensors and their relation to targeted temperatures. Moreover, Fig 9 depicts that the findings are less valid in the case of IVIFGs and it may takes lot of time to settle down the temperature ranges. On the other hand, if we utilize the domain of IVq-ROFGs for our application, then we can easily analyze all the temperature sensors by using $MSIS$. Furthermore, Fig 9 shows that the range of the FW of $MSIS$ in the domain of IVq-ROFGs is wide and flexible. In this regard, we can settle down properly a very high or very low temperature according to our need. Consequently, it is helpful to make an appropriate conclusions about temperatures on sensors and apply fuzzy logic to FACS in the paradigm of IVq-ROFGs.

**Table 13. The $MSIS$s for IVIFGs and IVq-ROFGs.**

| $MSIS$s | IVIFGs | IVq-ROFGs |
|---|---|---|
| $Z_1 = \{b, d\}$ | $\langle [0.9, 1.2], [0.5, 0.6] \rangle$ | $\langle [1.1, 1.6], [0.8, 1.4] \rangle$ |
| $Z_2 = \{c, e\}$ | $\langle [0.6, 1.0], [0.5, 0.9] \rangle$ | $\langle [0.7, 1.3], [1.0, 1.6] \rangle$ |

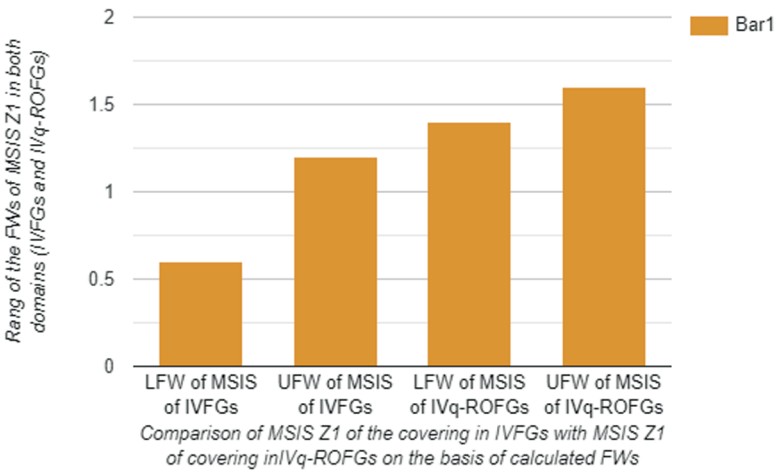

Comparison of MSIS Z1 of the covering in IVFGs with MSIS Z1 of covering inIVq-ROFGs on the basis of calculated FWs

**Fig 9. Comparison of MSIS Z1 of the covering in IVFGs with MSIS Z1 of covering in IVq-ROFGs on the basis of calculated FWs.**

**Table 14. The characteristic comparison of IVq-ROFGs with IVFGs, IVIFGs and IVPyFGs.**

| Characteristics | IVFGs | IVIFGs | IVPyFGs | IVq-ROFGs |
|---|---|---|---|---|
| Representation | Using only degree of membership | Using degree of membership and degree of non-membership | Using degree of membership and degree of non-membership | Using degree of membership and degree of non-membership |
| Effectiveness in clusters's engagement | Limited view, dealing only with degree of membership | Balanced view, dealing with degree of membership and degree of non-membership | Better view, dealing with degree of membership and degree of non-membership | Comprehensive view, dealing with degree of membership and degree of non-membership |
| Competency in dealing with real-world problems | Limited competency because dealing only with degree of membership | Better competency because dealing with degree of membership and degree of non-membership | Much better competent because dealing with degree of membership and degree of non-membership | highly competent because dealing with degree of membership and degree of non-membership |
| Adaptability in representation of maximal SISs. | Limited adaptability | adaptable | More adaptable | Highly adaptable |

# 7 Conclusion

There are several real-world uncertain problems that cannot be handled through FGs, IFGs and IVPyFGs, but can be explain easily through IVq-ROFGs due to its flexible nature. IVq-ROFGs offers more adaptability, accuracy and precision for the problems with uncertainties having two attribute yes or no. In this study, we have investigated the concepts of covering and matching based on SAs within the framework of IVq-ROFGs. We have added several novel terms like SM, SAC, SNC, SISs, PSM and produced many new results in the domain of IVq-ROFGs. Moreover, we have also addressed some special IVq-ROFGs like complete

IVq-ROFGs, complete bipartite IVq-ROFGs etc. Overall, we have extended the concepts presented for IFGs, PyFGs etc. To show the usefulness of this work, an application of covering in IVq-ROFGs in the analysis of FACS based on temperature sensors is also provided. We have observed that applying covering in IVq-ROFG to temperature sensors offers a more flexible and adaptable framework for FACS, simplifying the management of indoor temperature control. This study also opens the door to extending these concepts to fuzzy soft and neutrosophic graphs to address other real-world problems.

## 7.1 Implications of our proposed study

Our proposed notion of covering and matching in interval-valued q-rung orthopair fuzzy graphs has far-reaching implications some of them are stated belo. ering innovation in various fields:

**Theoretical implications**

1. Addition of the concepts of matching and covering in interval-valued q-rung orthopair fuzzy graphs.
2. Interconnections with other fuzzy structures such as intuitionistic fuzzy graphs, Pythgorean fuzzy graphs etc etc.
3. Generalizations of existing concepts of covering and matching in interval-valued intuitionistic fuzzy graphs.

**Practical applications**

1. Enhanced the model for the FACS setup, making it adaptable for other models with uncertain and imprecise data.
2. Improved optimization techniques for real-world problems such as resource allocation.
3. Effective analysis of interconnected phenomenon.

**Impact on fuzzy set theory and its applications**

1. Expansion of fuzzy set theory to accommodate interval-valued q-rung orthopair fuzzy sets.
2. Development of novel fuzzy logical methods.
3. Applications in soft computing, artificial intelligence, and machine learning.

**Interdisciplinary research**

1. Integration with operations research, management sciences etc.
2. Collaboration with experts systems with uncertainties specifically in electronics and electrical instruments.
3. Development of novel applications in emerging fields in the domain of fuzzy logic.

**Educational and industrial impact**

1. Structural enhancement of devices by incorporating fuzzy logic.
2. Adoption of interval-valued q-rung orthopair fuzzy graph-based methods in commercial and industrial settings.

## 8 Research limitations and future directions

Our study paves the way for significant advancements in interval-valued q-rung orthopair fuzzy graph theory and its applications, driving scientific progress and innovation in the years to come. In this work, we have presented the model based on limited temperature sensors. However, if we apply our model to large scale, it could be reformed by designing a particular algorithm to achieve the best solution. In this context, since the notion of interval-valued q-rung orthopair picture fuzzy graphs (IVq-ROPFGs) is not introduced in literature, one could provide comparatively more precise solution of the addressed problem by introducing the concepts of IVq-ROPFGs. Moreover, the algorithm presented in this study can be developed toward large-scale IVq-ROPFGs and the problems existing in several fields like block-chain, cyber-security, data science etc can be addressed.

## Author contributions

**Conceptualization:** Waheed Ahmad Khan, Sagheer Abbas, Akhlaq Ahmed, Madhumangal Pal, Muhammad Asif, Muhammad Saeed Khan.

**Formal analysis:** Waheed Ahmad Khan, Sagheer Abbas, Akhlaq Ahmed, Madhumangal Pal, Muhammad Asif.

**Investigation:** Waheed Ahmad Khan, Akhlaq Ahmed, Madhumangal Pal, Muhammad Saeed Khan.

**Methodology:** Waheed Ahmad Khan, Sagheer Abbas, Akhlaq Ahmed, Madhumangal Pal, Muhammad Asif, Muhammad Saeed Khan.

**Supervision:** Waheed Ahmad Khan.

**Validation:** Muhammad Saeed Khan.

**Visualization:** Muhammad Asif.

**Writing – original draft:** Waheed Ahmad Khan, Sagheer Abbas, Akhlaq Ahmed.

**Writing – review & editing:** Sagheer Abbas, Madhumangal Pal, Muhammad Asif, Muhammad Saeed Khan.

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
