## [Decision Letter · Decision Letter 0]

22 Sep 2024

PONE-D-24-37248Some Novel Concepts of Interval-valued q-rung orthopair Fuzzy Graphs and Computational Framework of Fuzzy Air

Conditioning SystemPLOS ONE

Dear Dr. Khan,

Thank you for submitting your manuscript to PLOS ONE. After careful consideration, we feel that it has merit but does not fully meet PLOS ONE’s publication criteria as it currently stands. Therefore, we invite you to submit a revised version of the manuscript that addresses the points raised during the review process.

We look forward to receiving your revised manuscript.

Kind regards,

Tareq Al-shami

Academic Editor

PLOS ONE

Journal Requirements:

3. Please ensure that you refer to Figure 1 in your text as, if accepted, production will need this reference to link the reader to the figure.

4. Please upload a copy of Figure 9, to which you refer in your text on page 21. If the figure is no longer to be included as part of the submission please remove all reference to it within the text.

5. We note you have included a table to which you do not refer in the text of your manuscript. Please ensure that you refer to Table 1 in your text; if accepted, production will need this reference to link the reader to the Table.

Additional Editor Comments:

Dear authors,

Take the following revisions beside reviewers' comments when you revise your manuscript:

1) The way of presenting symbols should be carefully updated, there are some typos, the way of writing PyFS in Definition 5 should be rectified.

2) The first two lines of the proof of Theorem 26 should be revised.

3) Refer to the unique contributions of this work

4) The conclusion section is primitive, neither deep investigation of the obtained results nor refer for future work

Reviewers' comments:

Reviewer's Responses to Questions

**Comments to the Author**

1. Is the manuscript technically sound, and do the data support the conclusions?

Reviewer #1: Yes

Reviewer #2: Yes

2. Has the statistical analysis been performed appropriately and rigorously? 

Reviewer #1: Yes

Reviewer #2: Yes

3. Have the authors made all data underlying the findings in their manuscript fully available?

Reviewer #1: Yes

Reviewer #2: No

4. Is the manuscript presented in an intelligible fashion and written in standard English?

Reviewer #1: No

Reviewer #2: Yes

5. Review Comments to the Author

Reviewer #1: 1)The quality of the English language should be improved carefully to enhance readability, such as the phrase "Suppose that G be a complete"

2)Please, justify the proposed concepts with an illustrative example since one can discuss the proposed application using q-rung orthopair Fuzzy Graphs

3) Recently, it was proposed new techniques to deal with membership and non0membership functions by changing their power such as (2,1)-fuzzy sets; (m,n)-Fuzzy sets; and (a,b)-Fuzzy soft sets. Investigate this progression in the introduction section and the possibility of future applications in the conclusion sections.

4) The values of Table 2a should be revised carefully and made the appropriate corrections

Reviewer #2: Dear authors

the abstract lacks clarity, focus, and a clear contribution to the field. First, the introduction of "fuzzy sets" is vague and poorly contextualized; it reads more like a generic description rather than an insightful introduction to the problem. The mention of interval-valued q-rung orthopair fuzzy sets (q-ROFSs) is abrupt and doesn't build on the previous information, leaving the reader confused about how it significantly improves over existing methods. The abstract does not clearly justify why q-ROFSs are a "more flexible tool," which is a critical oversight in demonstrating the novelty of the study.Furthermore, the objectives of the study are presented in a fragmented and confusing manner. The transition from introducing fuzzy graphs to the application in a fuzzy air conditioning system (FACS) lacks logical flow, making it hard to follow the progression of ideas. The abstract feels like a collection of loosely related concepts that are not tied together well.The second objective, providing a computational framework for the FACS, is extremely underdeveloped. There's no clear explanation of how this framework works, its unique contributions, or how it advances the field. Phrases like "provide evidence through comparative study" are vague and unsubstantiated, as the abstract doesn't clarify what exactly was compared or what the benchmark for success was.the abstract lacks any mention of managerial, practical, which would make the research more valuable and applicable. The overall tone is overly technical without being informative, making it inaccessible to a broader audience. The writing is overly complex without justification, leading to unnecessary confusion. This abstract needs a complete overhaul to ensure clear articulation of the problem, methodology, and the real-world relevance of the findings.

Incredibly, this manuscript lacks a dedicated section for the literature review, which raises the question: wouldn’t it be more appropriate to include a thorough review of relevant research? This would provide readers with a better understanding of the existing knowledge base and where your work fits in. Furthermore, there is no discussion section, which is crucial for comparing your findings with those of other studies. Readers are interested in seeing a real discussion between your results and previous research.Additionally, the conclusion is extremely brief. It could benefit from being divided into distinct sub-sections, such as theoretical contribution, managerial implications, and practical applications. As a decision-maker in a public or private sector organization, how would this research help me? Please clearly articulate these aspects.Moreover, include a section titled "Implications" that addresses the potential impact of this research on the scientific community in the coming months and years. A section on research limitations and future recommendations is also necessary for a well-rounded paper.

While the paper currently has structural and methodological shortcomings, I am inclined to give it a chance due to its contribution to advancing fuzzy concepts. With substantial revisions, it has the potential to meet the required standards.

6. PLOS authors have the option to publish the peer review history of their article (what does this mean?). If published, this will include your full peer review and any attached files.

Reviewer #1: No

Reviewer #2: No

---

## [Author Response · Author response to Decision Letter 1]

4 Nov 2024

Journal Requirements with Responses

Manuscript ID: PONE-D-24-37248

Title: Some Novel Concepts of Interval-valued q-rung orthopair Fuzzy Graphs and Computational Framework of Fuzzy Air Conditioning System

We appreciate the Journal Editor for their time and precious comments. No doubt, after endorsing these comments our article would become more effective. The point by point responses of the editor’s comments is as follows.

Comment #1. Please ensure that your manuscript meets PLOS ONE's style requirements, including those for file naming. The PLOS ONE style templates can be found at https://journals.plos.org/plosone/s/file?id=wjVg/PLOSOne_formatting_sample_main_body.pdf and https://journals.plos.org/plosone/s/file?id=ba62/PLOSOne_formatting_sample_title_authors_affiliations.pdf

Response. We have adjusted the authors affiliations and body of the manuscript according to the journal’s need.

Comment #2. Please provide a complete Data Availability Statement in the submission form, ensuring you include all necessary access information or a reason for why you are unable to make your data freely accessible. If your research concerns only data provided within your submission, please write "All data are in the manuscript and/or supporting information files" as your Data Availability Statement.

Response. The data availability statement is now rectified as “All relevant data are within the manuscript.”

Comment #3. Please ensure that you refer to Figure 1 in your text as, if accepted, production will need this reference to link the reader to the figure.

Response. The Figure 1 mentioned by the reviewer is actually Figure 2, it was a typo error and we have fixed it.

Comment # 4. Please upload a copy of Figure 9, to which you refer in your text on page 21. If the figure is no longer to be included as part of the submission please remove all reference to it within the text.

Response. Thank you very much to the reviewer, there was a typo mistake in the numbering of Figure 9 mentioned on page 21, it is actually a Figure 7, we have fixed all the typo errors throughout the manuscript, and adjusted the references of all the Figures and Tables, accurately.

Comment # 5. We note you have included a table to which you do not refer in the text of your manuscript. Please ensure that you refer to Table 1 in your text; if accepted, production will need this reference to link the reader to the Table.

Response. As per reviewer’s suggestions, we have referred the Table 1 in literature. In addition, we have fixed the references of all the Figures and Tables, carefully.

Editor’s Comments with Responses:

Manuscript ID: PONE-D-24-37248

Title: Some Novel Concepts of Interval-valued q-rung orthopair Fuzzy Graphs and Computational Framework of Fuzzy Air Conditioning System

We appreciate the Editor for their time and precious comments. No doubt, after endorsing these comments our article would become more effective. The point by point responses of the editor’s comments is as follows.

Dear authors,

Take the following revisions beside reviewers' comments when you revise your manuscript:

Comment #1. The way of presenting symbols should be carefully updated, there are some typos, the way of writing PyFS in Definition 5 should be rectified.

Response. As per reviewer’s suggestion, we have rectified the way of writing PyFSs in Definition 5.

Comment # 2. The first two lines of the proof of Theorem 26 should be revised.

Response. We have revised the first two lines of the proof of Theorem 26.

Comment # 3. Refer to the unique contributions of this work.

Response. We have addressed the Unique contributions of this work under the heading “Novelty of our work”.

Comment # 4. The conclusion section is primitive, neither deep investigation of the obtained results nor refer for future work.

Response. As per reviewer’s suggestion, we have revised the conclusion section and prescribed the future directions.

Reviewer-1 Comments with Responses:

Manuscript ID: PONE-D-24-37248

Title: Some Novel Concepts of Interval-valued q-rung orthopair Fuzzy Graphs and Computational Framework of Fuzzy Air Conditioning System

We appreciate the reviewer for their time and precious comments. No doubt, after endorsing these comments our article would become more effective. The point by point response of the reviewer comments are as follows.

Comment #1. The quality of the English language should be improved carefully to enhance readability, such as the phrase "Suppose that G be a complete".

Response. Thank you very much for the reviewer. We have revised the statements and replaced the phrase “Suppose that G be a complete” with “Let G be a complete”. In addition, we revised the statements that having such types of statements. We also carefully revised the manuscript to enhance the quality of English language.

Comment #2. Please, justify the proposed concepts with an illustrative example since one can discuss the proposed application using q-rung orthopair Fuzzy Graphs.

Response. As per reviewer’s suggestion, we have elaborated all the introduced notions with illustrative Examples, i.e., Example 34, 35, 45, 46 and 48.

Comment #3. Recently, it was proposed new techniques to deal with membership and non0membership functions by changing their power such as (2,1)-fuzzy sets; (m,n)-Fuzzy sets; and (a,b)-Fuzzy soft sets. Investigate this progression in the introduction section and the possibility of future applications in the conclusion sections.

Response. As per reviewer’s suggestion, we have cited the newly published literature related to the generalizations of fuzzy sets in the introduction section. In conclusion section, we have provided a possible future application of this work.

Comment #4. The values of Table 2a should be revised carefully and made the appropriate corrections

Response. As per reviewer’s suggestion, the values of Table 2a are rectified and we have also revised the values of Tables 2b, 4 and Figures 2, 4.

Reviewer-2 Comments with Responses:

Manuscript ID: PONE-D-24-37248

Title: Some Novel Concepts of Interval-valued q-rung orthopair Fuzzy Graphs and Computational Framework of Fuzzy Air Conditioning System

We appreciate the reviewer for their time and precious comments. No doubt, after endorsing these comments our article would become more effective. The point by point response of the reviewer comments are as follows.

Reviewer #2:

Dear authors,

(1) The abstract lacks clarity, focus, and a clear contribution to the field. First, the introduction of "fuzzy sets" is vague and poorly contextualized; it reads more like a generic description rather than an insightful introduction to the problem. The mention of interval-valued q-rung orthopair fuzzy sets (q-ROFSs) is abrupt and doesn't build on the previous information, leaving the reader confused about how it significantly improves over existing methods. The abstract does not clearly justify why q-ROFSs are a "more flexible tool," which is a critical oversight in demonstrating the novelty of the study. Furthermore, the objectives of the study are presented in a fragmented and confusing manner. The transition from introducing fuzzy graphs to the application in a fuzzy air conditioning system (FACS) lacks logical flow, making it hard to follow the progression of ideas. The abstract feels like a collection of loosely related concepts that are not tied together well. The second objective, providing a computational framework for the FACS, is extremely underdeveloped. There's no clear explanation of how this framework works, its unique contributions, or how it advances the field. Phrases like "provide evidence through comparative study" are vague and unsubstantiated, as the abstract doesn't clarify what exactly was compared or what the benchmark for success was? the abstract lacks any mention of managerial, practical, which would make the research more valuable and applicable. The overall tone is overly technical without being informative, making it inaccessible to a broader audience. The writing is overly complex without justification, leading to unnecessary confusion. This abstract needs a complete overhaul to ensure clear articulation of the problem, methodology, and the real-world relevance of the findings.

Response. Thank you very much for the reviewer for his careful revision and valuable suggestions. We must describe the incorporations in the following points:

i. We have revised the abstract section, properly. In this regard, we have elaborated the significance of study on q-ROFSs compared to other generalizations of FSs.

ii. We have updated the section of motivation and novelty of our work.

iii. We also updated the application section in order to make a logical flow. In this regard, we relate a FACS with temperature sensors, instead.

iv. As per reviewer’s suggestion, we have provided a pseudocode and algorithm which indicate how this computational framework works.

v. We have also rectified the section of comparative analysis by elaborating the worthiness of our study through examples. Moreover, we have also added a subsection “Discussions”, in which we compare our proposed study by existing ones.

(2) Incredibly, this manuscript lacks a dedicated section for the literature review, which raises the question: wouldn’t it be more appropriate to include a thorough review of relevant research? This would provide readers with a better understanding of the existing knowledge base and where your work fits in.

Response. As per reviewer’s suggestion, we have added a section “Literature review” in which we have revised the relevant study and clearly highlighted the importance of our work.

(3) Furthermore, there is no discussion section, which is crucial for comparing your findings with those of other studies. Readers are interested in seeing a real discussion between your results and previous research.

Response. We have added a discussion as a subsection of a Section 6 which includes the comparison between our findings to the existing studies.

Additionally, the conclusion is extremely brief. It could benefit from being divided into distinct sub-sections, such as theoretical contribution, managerial implications, and practical applications. As a decision-maker in a public or private sector organization, how would this research help me? Please clearly articulate these aspects. Moreover, include a section titled "Implications" that addresses the potential impact of this research on the scientific community in the coming months and years.

Response. As per reviewer’s suggestion, we have revised the conclusion section. We have elaborated the notions that we have introduced. Moreover, we have also provided a theoretical contributions, managerial implications and practical applications of our study in the subsection " Implications of our proposed study”.

(4) A section on research limitations and future recommendations is also necessary for a well-rounded paper.

Response. As per reviewer’s suggestion, we have added a section about the research limitations and future directions of our proposed study.

While the paper currently has structural and methodological shortcomings, I am inclined to give it a chance due to its contribution to advancing fuzzy concepts. With substantial revisions, it has the potential to meet the required standards.

---

## [Decision Letter · Decision Letter 1]

3 Jan 2025

Some Novel Concepts of Interval-valued q-rung orthopair Fuzzy Graphs and Computational Framework of Fuzzy Air

Conditioning System

PONE-D-24-37248R1

Dear Dr. Khan,

We’re pleased to inform you that your manuscript has been judged scientifically suitable for publication and will be formally accepted for publication once it meets all outstanding technical requirements.

Kind regards,

Tareq Al-shami

Academic Editor

PLOS ONE

Additional Editor Comments (optional):

The suggestions and comments given by the reviewers have been made. The manuscript may now be accepted.

Reviewers' comments:

Reviewer's Responses to Questions

**Comments to the Author**

1. If the authors have adequately addressed your comments raised in a previous round of review and you feel that this manuscript is now acceptable for publication, you may indicate that here to bypass the “Comments to the Author” section, enter your conflict of interest statement in the “Confidential to Editor” section, and submit your "Accept" recommendation.

Reviewer #2: (No Response)

2. Is the manuscript technically sound, and do the data support the conclusions?

Reviewer #2: Yes

3. Has the statistical analysis been performed appropriately and rigorously? 

Reviewer #2: Yes

4. Have the authors made all data underlying the findings in their manuscript fully available?

Reviewer #2: (No Response)

5. Is the manuscript presented in an intelligible fashion and written in standard English?

Reviewer #2: No

6. Review Comments to the Author

Reviewer #2: (No Response)

7. PLOS authors have the option to publish the peer review history of their article (what does this mean?). If published, this will include your full peer review and any attached files.

Reviewer #2: No

---

## [Editor Report · Acceptance letter]

PONE-D-24-37248R1

PLOS ONE

Dear Dr. Khan,

I'm pleased to inform you that your manuscript has been deemed suitable for publication in PLOS ONE. Congratulations! Your manuscript is now being handed over to our production team.

Kind regards,

on behalf of

Professor Tareq Al-shami

Academic Editor

PLOS ONE